# E-Government 3.0: An AI Model to Use for Enhanced Local Democracies

Catalin Vrabie 🔗

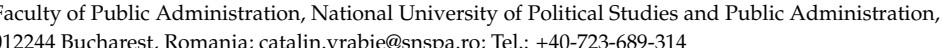

Faculty of Public Administration, National University of Political Studies and Public Administration, 012244 Bucharest, Romania; catalin.vrabie@snspa.ro; Tel.: +40-723-689-314

**Abstract:** While e-government (referring here to the first generation of e-government) was just the simple manner of delivering public services via electronic means, e-gov 2.0 refers to the use of social media and Web 2.0 technologies in government operations and public service delivery. However, the use of the term 'e-government 2.0' is becoming less common as the focus shifts towards broader digital transformation initiatives that may include AI technologies, among others, such as blockchain, virtual reality, and augmented reality. In this study, we present the relatively new concept of e-government 3.0, which is built upon the principles of e-government 2.0 but refers to the use of emerging technologies (e.g., artificial intelligence) to transform the delivery of public services and improve governance. The study objective is to explore the potential of e-government 3.0 to enhance citizen participation, improve public service delivery, and increase responsiveness and compliance of administrative systems in relation to citizens by integrating emerging technologies into government operations using as a background the evolution of e-government over time. The paper analyzes the challenges faced by municipalities in responding to citizen petitions, which are a core application of local democracies. The author starts by presenting an example of an e-petition system (as in use today) and analyses anonymized data of a text corpus of petitions directed to one of the Romania municipalities. He will propose an AI model able to deal faster and more accurately with the increased number of inputs, trying to promote it to municipalities who, for some reason, are still reluctant to implement AI in their operations. The conclusions will suggest that it may be more effective to focus on improving new algorithms rather than solely on 'old' technologies.

**Keywords:** machine learning; petitions; G2C; governance; innovation; citizen participation; artificial intelligence; natural language processing

---





## 1. Introduction

The rapid adoption of digital technologies in public service delivery and governance has transformed the way governments operate, interact with citizens, and deliver public services. The evolution of e-government has progressed from basic electronic delivery of services (e-government 1.0) to the use of social media and Web 2.0 technologies (e-government 2.0), resulting in significant changes in the roles of citizens and governments in public service delivery [1,2].

However, the emergence of new technologies such as artificial intelligence (AI), blockchain, and the Internet of Things (IoT) has paved the way for a new era of e-government, referred to as e-government 3.0 [3–6]. This new version of the concept holds the promise of transforming public service delivery and governance by integrating emerging technologies into government operations [7,8].

The article will explore the potential of e-government 3.0 to enhance citizen participation, improve public service delivery, and increase responsiveness and compliance of administrative systems in relation to citizens. The author will start its analysis based on empirical evidence from a case study on the Brasov municipality, Brasov is considered one of the smartest Romanian cities as of today [9–11], to assess the impact of e-government 3.0 on

a particular aspect of governance and public administration: e-petitioning. Ultimately, this paper seeks to contribute to the ongoing scientific debate on the future of e-government and the potential of emerging technologies to transform public service delivery and governance.

As stated in a few of the author's previous studies [12,13], while municipalities aim to enhance citizen participation and openness by developing online platforms for communication, one of the widely used tools is e-petitioning, which is utilized by both local and central governments. Although social media networks have also been used by public organizations to engage with their constituents and receive feedback, most administrative courts do not consider social media discussions as legal actions, unlike e-petitions, which do have official status. Although AI can assist in addressing citizens' concerns on social media also, e-petitioning should be regarded as the preferred AI-powered solution for resolving administrative issues.

The role of urban computing in sustainable smart cities, outlining recent developments, use cases, and research challenges in the field were addressed by Hashem et al. [14], while Zhao et al. investigates the influence of digital and technological advancement on sustainable economic growth and analyzes the impact of variables such as E-government Development Index (EGDI), Internet Users' (IU) growth, and information and communications technology (ICT) exports [15]. Both articles highlight the importance of harnessing technological advancements to achieve sustainable development and provide insights into the opportunities and challenges of doing so. Modern citizens demand prompt, efficient, and high-quality services from their public authorities, especially since trust in governments and their services has been diminishing worldwide [16,17].

Consequently, citizens demand better infrastructure, improved services, and adaptive leadership. However, due to increasing demands and constrained public budgets, effective solutions are often delayed, and administrative capacities may be lacking [18]. Therefore, the literature on public management suggests that AI applications can play a crucial role in generating and sustaining good governance by mitigating these challenges, such as long delays, unskilled personnel, and overall administrative inefficiencies.

The study hypothesis is that e-government 3.0 has the potential to increase the responsiveness of administrative systems, thus enhancing citizen participation. The author will explore this hypothesis through a synthetic case study proposing an automated text analysis method over the e-petition systems in use today. Additionally, the study hypothesizes that AI applications can play a crucial role in generating and sustaining good governance by mitigating challenges such as long delays, unskilled personnel, and administrative inefficiencies.

The study objective is to explore the potential of e-government 3.0 to enhance citizen participation, improve public service delivery, and increase responsiveness and compliance of administrative systems in relation to citizens by integrating emerging technologies into government operations.

After the introductory section, the article will feature a significant number of studies, articles, and analyses with the purpose of linking the field of governmental studies to the rapidly evolving field of artificial intelligence. In the third section, the reader will be guided from the scholarly research to the dataset that the author intends to utilize to substantiate the hypothesis. Additionally, Section 4 of the article will introduce a machine learning (ML) model that is trained and validated on a set of data obtained from one of Romania's smart cities (Brasov). This section will include dedicated subsections that will explain the behavior of the model and the expected outputs in a lightly technical manner. To achieve this, the author will start by examining past successes in machine learning that were used to validate optimistic views regarding the future of e-government 3.0. The findings presented in Section 5 and the subsequent discussion in Section 6 will validate the assumption that AI technologies are necessary for the proper development of government-to-citizen (G2C) interaction. The author's vision for the use of AI, research limitations, and future work will also be outlined in Section 6. Finally, the article concludes with the last section.

## 2. Literature Review

Modern citizens expect prompt, effective, and high-quality services from their public authorities, and the decline in trust in governments and their services is a worldwide phenomenon [16,17,19]. This has led to a growing demand for better infrastructure, improved services, and adaptive leadership. However, limited public budgets and increasing demands create serious constraints for meeting these expectations, leading to delays in presenting effective solutions, under-skilled personnel, and overall poor administrative capacities [20].

Technological advances offer solutions to both businesses and governments, and integration of artificial intelligence has the potential to positively impact global productivity and environmental outcomes, and the development of sustainable business models is necessary [21].

The literature on public management suggests that AI applications can address these challenges and help generate and sustain good governance [22–24]. For example, AI can improve public service delivery by enhancing the quality and efficiency of services [24,25], automating administrative tasks [26], and supporting decision-making processes [27,28]. Moreover, AI can enhance transparency and accountability, as well as increase citizen participation and engagement [29,30]. Ibtissem et al. used advanced statistical methods to investigate the challenges faced by emerging economies in addressing issues of poor governance in public services [31]. Thus, AI has the potential to transform public management and governance, helping public authorities to better meet citizens' expectations and improve trust in government services.

AI can play a crucial role in e-petitioning by summarizing and triaging petitions, providing automated responses to routine queries [32,33], and identifying petitions that require further analysis from specialized departments [34,35]. It can assist in decision-making by providing evidence for a more comprehensive reply that is compliant with national or international regulations [36]. AI can filter petitions to verify their eligibility, compare subjects and frequencies, and measure organizational efficiency [37]. By performing these tasks, AI can save time, energy, and resources and limit redundancies and time waste [38]. It can also use the 'compare and comply' functions to navigate regulations and ensure that official replies are correct and complete. AI can identify urgencies in petitions' texts using sentiment analysis and trigger faster reactions from the government, increasing confidence in public authorities [39,40]. Learning and reasoning are also critical components to consider in utilizing AI in e-petitioning [41].

After reviewing the research outlined in *Sustainability* (issues 2020–2023), *Mathematics* (issues 2020–2023), *Government Information Quarterly* (issues 2020–2023), and *International Journal of Web Services Research* (issues 2020–2023), one can conclude that much of the focus is on e-government in general and little on the use of top technologies (AI, machine learning (ML), natural language processing (NLP), and robotic/intelligent process automation (RPA/IPA) technologies, seen here as top technologies) for improving governance processes.

As early as 1999, Jon M. Kleinberg from Cornell University [42] studied the network structure of a hyperlinked environment and developed a set of algorithmic tools for extracting information from the link structures of such environments. At the time, the study focused on a variety of contexts on the World Wide Web. Later, in 2011, Hreňo et al. [43] described the approach to semantic interoperability of e-government services. Piaggesi [44] researched the future of connectivity and provided a snapshot of Latin America, recommending that the role of government in providing universal service is very important for a proper transition to e-government 3.0. Verma [45] made a comprehensive bibliometric review of 353 research articles published between 2010 and 2021 to discern the performance of public servants. The author concluded that governance structures, together with the whole society, are becoming smarter by using smart technologies. However, by reading the text, one can admit that this is a projection of the author's hopes for the foreseeable future, but there is no clear indication of when this will happen.

A group comprising seven social and computer science specialists at McKinsey & Company created a chart in which they mapped the most encouraging technologies based on their potential applications in domains that could be beneficial to society. They relied on a study conducted in 2018 and concluded that the most valuable technologies are deep learning, natural language processing, image and video classification, object detection, and language understanding. All of these technologies are related to information verification and validation [46].

Moreover, Madan and Ashok [47], through a systematic literature review, identified contextual variables as factors influencing AI adoption, as discussed in the literature. The authors concluded that governance maturity is identified as an important component of managing AI implementation. Additionally, Ahn and Chen [48] explored the perception of public employees regarding the use of AI technologies in government. The authors found that government employees hold a positive view regarding the benefits and potential of AI technologies in the public sector, heaving high expectations over the integration of AI, believing it will enhance the efficiency and quality of government operations.

Kumari et al. [49] proposed techniques based on sentiment analysis meant to improve the performance of employees connected with users by different platforms. Furthermore, Lu et al. [50] focused on applying a cross-domain aspect-based sentiment analysis model to word embeddings.

Similarly, Yu et al. [51] proposed a model containing a sentence encoder together with a semantic and syntax learning module for sentiment classifier, which is considered important for the present study on citizen petitions. If implemented in e-petitioning systems of government 3.0, the actual state of web apps will greatly improve, and citizens will have a more streamlined and efficient way to engage with their government.

Eom, Lee, and Zankova [52,53], focusing on dilemmatic situations in which to use technologies, provided an overview of previous literature on digital government transformation, stating that governments, by adopting actor-based computing models, along with large-scale data, can enhance their ability to identify real-world complexity, discern patterns in data, and leverage them to enhance its actions. This, in turn, can result in cost savings and better anticipation of future events.

McKinsey & Company conducted a recent study [54] where they showed enthusiasm for Generative AI software that can display creativity, which was previously considered a trait exclusive to humans. Some of the applications of these tools align with the topic of this article, including writing, documenting, and reviewing texts, as well as extracting information from large amounts of legal documents and answering intricate questions.

Andrew Ng, a Stanford professor and co-founder of Coursera and Google Brain, in a keynote speech at the AI Frontiers conference, said [55]: 'About 100 years ago, electricity transformed every major industry. AI has advanced to the point where it has the power to transform every major sector in coming years'.

## 3. Materials and Methods

For the present study, officials from the Romanian city of Brasov agreed to supply anonymized data, which comprised 12,935 petitions directed to the municipality in the year 2022 (1 January 2022–31 December 2022) via multiple communication channels (e-mail, phone apps, instant messages, Web platform and by phone)—Appendix A.

Each of them was converted using 118 indicators, seen as vectors, labeled in 47 classes that are also seen as layers. Previously, the responsibility of carrying out this task fell on the city hall employees, seen as experts who dealt with petitions as part of their daily duties. During labeling, experts were also clustering data based on similar text content.

As sample data, for the inference phase, a number of 1295 petitions were taken into consideration. At the end of this process, therefore, before starting the analyses, the sample in use consisted of 152,810 items.

At this stage, the author is willing to mention that other criteria of analysis were also taken into consideration: marital status (if directly or indirectly disclosed by the

sender), a platform he/she used (mobile, laptop/PC), references to other documents such as legislation or norms. All those are considered extra information but are important for building the statistics.

On this data set, a cleaning operation was performed in order to fully anonymize the data–all of the petitions had names, emails phone numbers, or similar data that could link the content to the sender; therefore, a full set of indicators were dropped resulting in a total number of 151,515 valid inputs. The author wants to mention here that the dataset received for this experiment consisted in petitions that were already answered by the city hall employees; therefore, they were considered valid, and it was most probable that a live model would face multiple invalid inputs. There are more limitations in a dedicated section at the end of the article.

Cleaning operations also consisted of the following:

1. Tokenization: Split text into individual words or tokens to allow for further processing;
2. Removing punctuation;
3. Spell correction;
4. Removing URLs and HTML tags;
5. Removing special characters;
6. Removing emoticons;
7. Removing offensive and bad words.

For the analysis itself, Google Colab [56] was used for its free access to a machine learning environment that allows one to write and run Python code, including machine learning algorithms. Moreover, the platforms allow using pre-trained models from popular machine learning frameworks such as TensorFlow and PyTorch. For this experiment, the author was using TensorFlow alone as the development platform with adjusted open-source software such as BERT [57] for text analyses using Index-Based Encoding and Bag of Words (BoW) techniques [58,59] fed up with texts from the data set used for this experiment. For tabular data (obtained after the triage were the first couple of indicators, such as age, marital status, and activism) TabNet [60] was used. Visual representations for the article were reproduced with the help of Tensor Flow Playground [61].

Few words describing the city: Brasov is located in central Romania at a reasonably high altitude (with heavy snows in the winter, when people tend to comply more about the inefficiencies of public administration during this season) and is serving as the capital of its county. It boasts a population of roughly 238,000 residents [48] (about 1.24% of Romania's population and approximately eight times smaller than Bucharest, the country's capital) and is recognized for its strong commercial and industrial sectors, making its population a very active one, with an average age of about 42 years (less than the country average). The city is governed by both a mayor and a city council.

## 4. The AI Model Proposed

### 4.1. Related Works

In the legal field, AI excels in handling repetitive and routine tasks [62], as is the case with AI in general. One of the earliest and most notable applications of AI in law was in the discovery phase of a trial, specifically with document classification. The initial approach involved searching for keywords to automate this process, but this was flawed because an idea or concept can be expressed in various ways, and certain keywords may be missed. Eventually, machine learning (ML) and natural language processing (NLP) algorithms were used; teams of lawyers classified samples of documents, and then the algorithms analyzed the patterns of words and combinations to identify which documents were responsive to the request [63]. This saved a significant amount of time for future queries. However, the results were not a binary classification; instead, the algorithm produced a probability score of a document being responsive [64]. Those with a high probability score are turned over, while those with a low score are disregarded. The ones placed in between would require human review.

Since most petitions require legislative input for answering, a similar model can be used. According to the statistics made by Brasov city hall, petitions follow common themes that arise in various scenarios [65]. Considering this, the process of document classification is considered routine, given that the public servants are creating a protocol and repeatedly applying it. However, automating this process is a lengthy endeavor since the information being analyzed, whether a document was responsive or not, is presented in text format. Without a method for a computer to comprehend language, the routine aspect of the work could not be achieved. Now that language processing has advanced enough to enable this, the process can be smoothly executed.

Moreover, if recently developed NLP systems of Generative AI, such as ChatGPT [66,67], are put in place, answering petitions after a proper classification of legal documents, as mentioned above, will be just a 'compare and comply' routine tasks [12].

### 4.2. Input Layer

Initially, by the use of Authority and Hubs Distribution algorithms [42], the system evaluated the degree of association between words found in the subject lines of all the petitions in order to classify the citizens' requests. This involved scoring the strength of connections among the words.

As shown in Figure 1, the words in the subject lines (59 nodes, or unique words out of a total of 66 and 44 edges/connections between the nodes) are interlinked (the darker the links and dots are, the stronger the connection is), with no isolated words. However, the connections between them are still weak at this point but help in classifying the main text corpus.

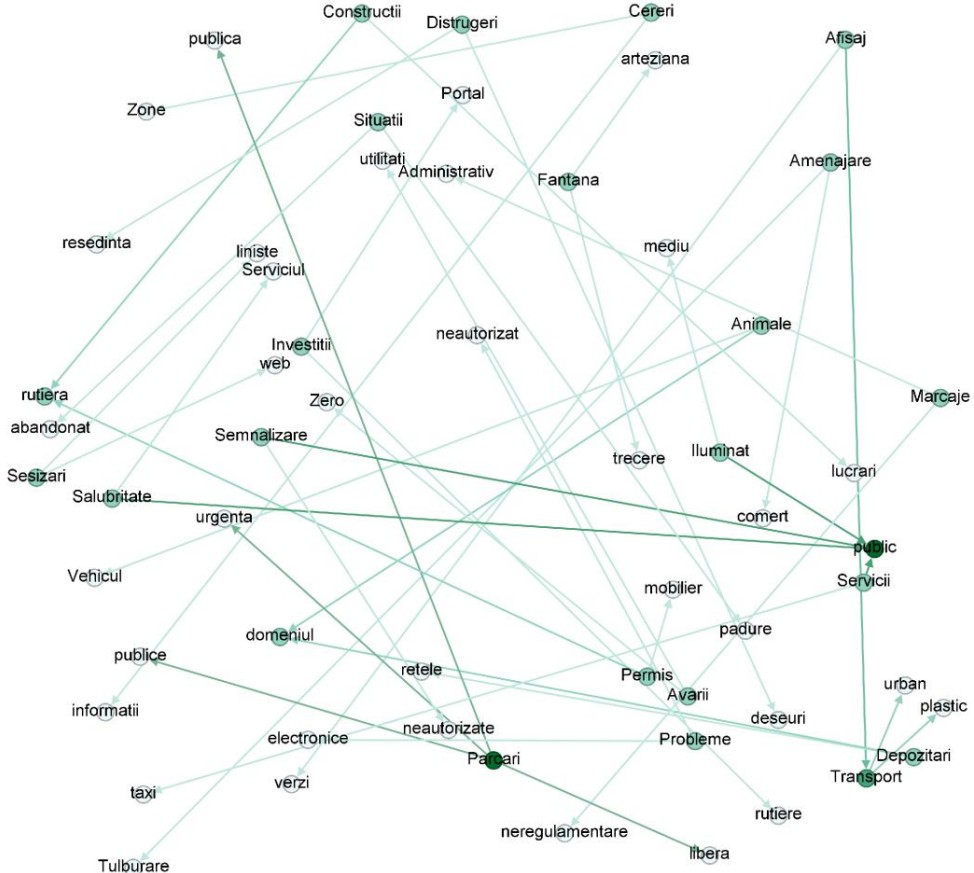

**Figure 1.** Analysis and visualization of keywords (in Romanian language) as they are found in the subject lines of petitions, first classification task.

### 4.3. Hidden Layers

The issue now is that the system does not know anything about the content of a petition or Facebook messages, so there is a strong need to translate it into a form that the computer can understand, and this is a feature vector. One possible way to perform this translation is to ask experts (public servants) about the content and concatenate the answers in a binary vector; this is what AI experts call supervised learning. Labeling content can be performed for each petition on the training set [68]. In fact, there is no need to allocate resources for labeling activities; the system can simply observe human actions and can analyze the patterns of words and combinations to properly label each petition.

Dealing with vectors and labels helps in translating the problem into a geometric form. If each one of those vectors is represented as a point in space, and if there is a corresponding label with those points, then the system may learn from the data.

In Figure 2, there are users of the system who sent a petition to the city hall (training set). On the bottom part, one can see petitions (set as a vector of importance) that could be treated with ease by the municipality, while on the upper part, there is an important issue that needs to be taken into consideration on a fast peace.

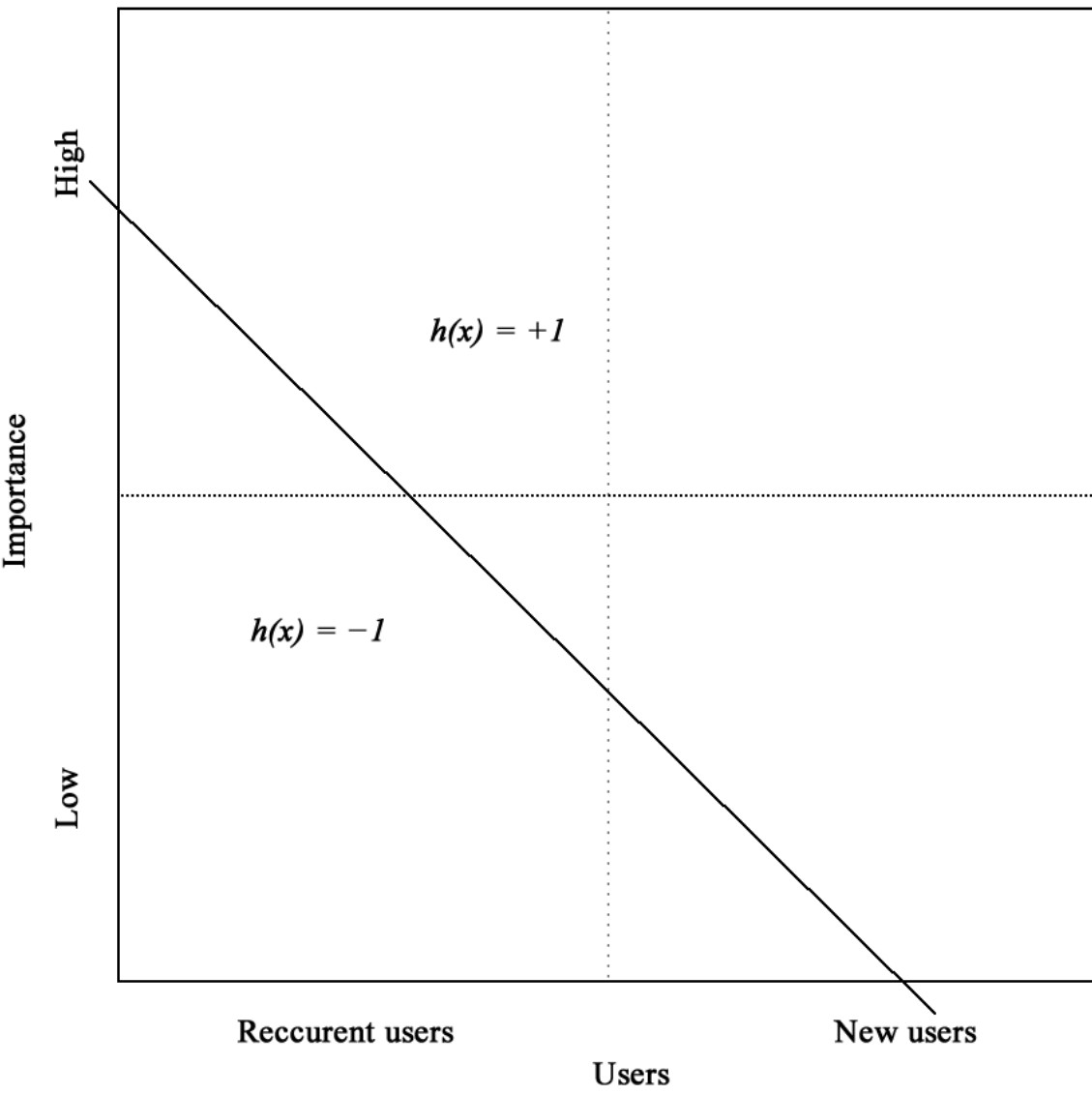

**Figure 2.** Generic model of predicting the importance (set as vector) of the topic addressed based on how active the citizens/users of the petitioning system are.

The line between left and right is the line the system is drawing on how likely the topic addressed is recurrent, which may already be answered for another citizen or even solved.

The slant line, however, represents the classifying function. For this article, the author generically defined it as *h(x)*.

However, Figure 2 represents just one vector of the model. For the purpose of this article, the author has chosen to present the 'importance' vector for a better understanding. There could be unlimited vectors grouped into an unlimited number of layers based on location, recurrence, reason (personal vs. general), and so on, with the 'importance' vector being just one of them. Overlapping all these different layers makes the system much more complex and, therefore, much more accurate in scoring the importance of precision (the ratio of positive predictions that are correct; those petitions from the upper right corner) and recall (the ratio of all positives that the final model is catching: the number of petitions misplaced by the system over the total number) [69].

### 4.4. Training Model

Figure 3 gives the visual representation of the training model. The model is elastic; it gives the possibility to be adjusted by the administrators by allowing them to add multiple layers and vectors.

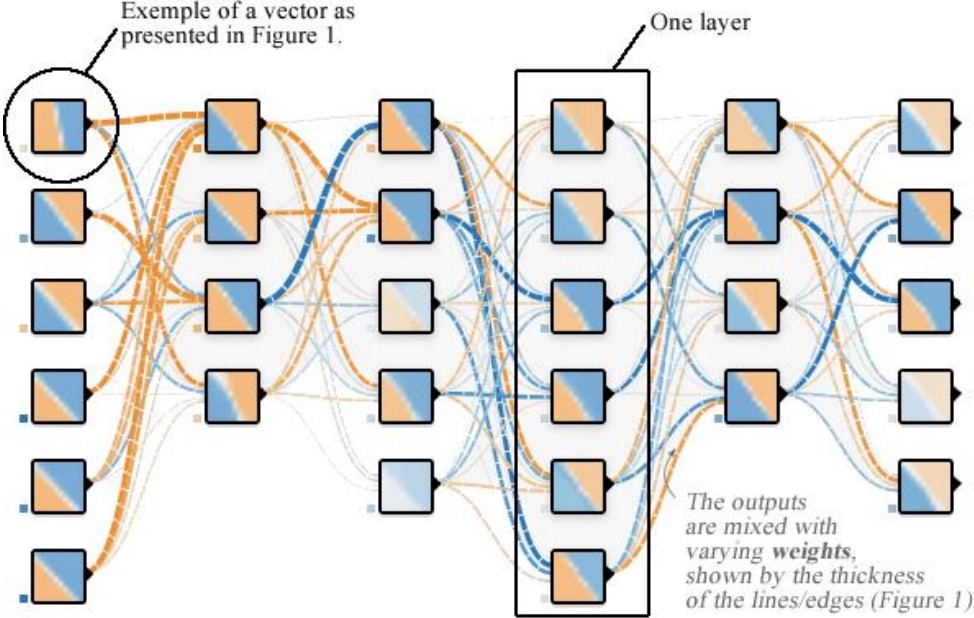

**Figure 3.** Training model (simplified); visual representation with Google TensorFlow (Source: github.com [61] – accessed on 19 March 2023).

Layers represent a group of interconnected items that, when computed together, helps the model perform better. Layers are based on features seen as vectors and considered important by the system administrators, such as language (e.g., using bad or offensive words), reason (e.g., already known malfunctions of different systems in the city: power supply, water linkage), geographical (e.g., areas that are confronted with the same problem, potholes for example, and different citizens keep sending messages with it), activities (e.g., music from a nearby festival), and the reliability of user (based on its previous posts on city hall social media official pages/petitions/messages, using sentiment analysis tools). In other words, the connection between vectors is made by measuring the weights of the edges (as seen in Figure 1).

Once the model is trained and compiled, the system assigns a score to each petition and performs specific actions based on a predetermined threshold.

When the system is faced with challenging texts (for example, questions addressed by citizens that demand intricate solutions that the system has not encountered before), the software generates multiple responses, each with a corresponding probability that signifies how confident it is about the correctness of the answer (e.g., 0.92, 0.84, 0.76–known as confidence numbers). The system administrators will determine a threshold or cut-off value that serves as a guideline for deciding which responses the machine can handle. Specifically, if the probability of a given response surpasses the threshold (e.g., 0.90), then the machine can take care of it and answer fully. However, if it falls below the threshold, the query needs to be addressed to a human operator.

### 4.5. How to Increase Precision and Recall

In general, the city hall has the identities of the citizens who address it by petitioning the places where they live. Therefore, it understands the problems they face and the problem they might complain about. It can then use this information to generate a list of profiles belonging to people who are frequent complainants (a large number of complainants tend to repeat their actions even if they receive a positive opinion from the city). In addition, it can use public information about active users on its official Facebook page. Correlating this information with sentiment analysis predictions, the system can be more efficient with improved effectiveness in assigning scores based on which it can consider certain actions.

It is interesting to mention here that the so-called Jevons Paradox [70–72], which states that improvements in efficiency and technology correlated with cost reduction, which initially leads to a decrease in resource use, may result in an overall increase in consumption/aggregate demand. As a result, the author predicts that the increased ease of use and speed of the system may lead to higher demand, offsetting any efficiency gains and placing greater pressure on administrative resources. This is mostly because the easiness of using the system, correlated with the speed at that the apps are answering/solving the issue, will encourage more use, leading to increased demand that may offset the savings gained from increased efficiency. This paradox highlights the need for a holistic approach to resource management that takes into account not just efficiency gains but also the behavior and citizens' responses to these gains.

### 4.6. Output Layer

Combating recall can be difficult, but since the robustness of a petition system should not be as strong as that of a financial one (e.g., dealing with fraud detection), such a system will definitely release the pressure from the public servants when dealing with large amounts of citizens complains just by acting as follows:

1.  It takes a soft action–backlog, by sending the petition for human investigation while helping with extracting relevant information from the legislative framework in order to help the public servant in giving an accurate answer to the complaint;
2.  It takes strong action, acting on behalf of humans (independently), generating narratives, and giving all necessary information to the citizen. It could also actively engage in a dialog using more advanced NLP capabilities (such as newly released GPT-4 [73]) if necessary;
3.  Pass action. In this scenario, the AI system could respond in a gentle manner, using language and phrases intended to de-escalate any potential argument with a confrontational citizen.

In any of the above cases, the public servants will receive a lot of help from such a system, while citizens will also receive more trust in the local government, knowing that the officials are active in solving their problems. If we sum up all the options above, we can see the efficiency of the system. Moreover, software bots can be used to gather information on public perceptions of various actions of officials or different agencies and, based on sentiment analysis, can study public mood and give clotted feedback to the municipality in order for it to improve its services [74,75].

For a better understanding, Figure 4. provides a visual representation of the model pipeline. In the real world, however, the data might not be as well balanced as they are in the presented outcome.

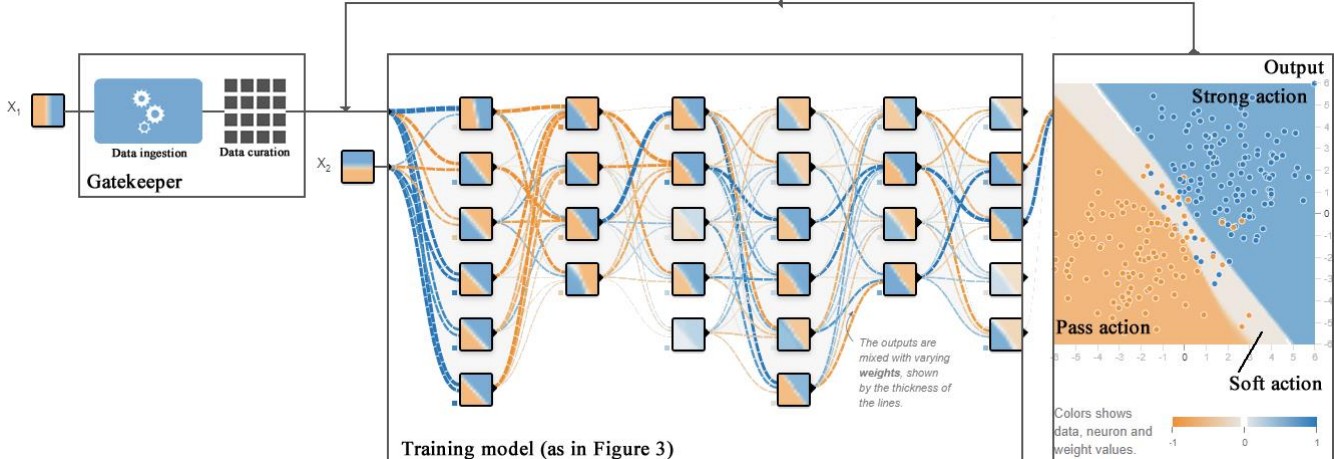

**Figure 4.** Model pipeline (visual representation with Google TensorFlow (Source: github.com [61]).

$X_1$ and $X_2$ are to be seen, for the purposes of this paper, as data inputs, such as $X_1$ from petitions and $X_2$ from messages posted on the official Facebook page of the institution.

Data curation refers to the process of cleaning input data (as mentioned in Section 3) for use in the model, making it relevant and reliable. In this experiment, the author replaced abbreviated words with their meaning together by converting them into vectors (numerical form) by Index-Based Encoding and Bag of Words (BoW) techniques [58,59].

The results of human investigation, which are considered as 'soft action', will be fed back into the system after the output phase. This feedback is aimed at readjusting the vectors to improve accuracy for the next input.

## 5. Results

In order to test the model, the author took real data from one of the big Romanian cities, Brasov, as described in the Materials and Methods section, and fed them into the system.

A full set of connections (made on 10% full texts from the training set at the inference phase) are to be seen in Figure 4.

Table 1 is a sample of the data set used for training.

**Table 1.** Sample of the dataset used for training.

| ID | Item | Value | Observations/Details |
|---|---|---|---|
| I1 | Gender * | 0/1/2 | 0—not known/1—man/2—women |
| I2 | Age group ** | 0 to 6 | 0—not known/6 > 70 |
| I3 | If it is on behalf of a company/firm | 0/1 | 0—ns/1—no/2—yes |
| I41 | Geographical 1 *** | 0 to 88 | divided in 8 major subgroups, each redivided into another 8 subgroups (0 for undisclosed) |
| I42 | Geographical 2 | 0/1/2 | 0—ns/1—the sender is living in a block of apartments/ 2—in a house (with land/garden) |

**Table 1.** *Cont.*

| ID | Item | Value | Observations/Details |
|---|---|---|---|
| I5 | Type of petition | 0/1/2/3/4/5 | 0—ns/1—demand/2—complain/ 3—referral/4—audience/5—proposal |
| I6 | Attachment | 0/1 | no/yes |
| I7 | Subject of petition | 0 to 9 | based on the words written in the Subject field (different from I3); 0 for ns |
| I81 | Active **** | 0/1/2 | 0—first/1—second/2—multiple |
| I82 | Active on official social media page | 0/1/2 | 0—first/1—second/2—multiple |
| I91 | Content 1 | 0/1 | if it refers to a neighbor/s (as a specific person/s) |
| I92 | | 0/1 | if it refers to the neighborhood |
| I101 | Content 2 | 0/1/2 | 0—no/1—if is regarding parking (in connection with I81)/2—if it regards parking (in connection with I82) |
| I102 | | 0/1 | if it regards public utilities (in connection with the I82) |
| I11 | Content 3 | 0/1 | 0—no/1—the content refers to the sender's own facilities (in connection with I32) |
| I12 | [ … ] ***** | [ … ] | [ … ] |

* extracted from the First name (in Romanian language, the vast majority of First names that end with 'a' belong to women); ** if directly (mentioning it in plain text) or indirectly (mentioning he/she is a student or a retired person, etc.) disclosed by sender; *** based on the address; **** if the person dropped more than one petition; ***** as mentioned earlier, there are several additional indicators that follow.

Below, in Table 2, one can see the results extracted for the purpose of interpretability, as given by the machine based on the inputs presented in Table 1.

**Table 2.** Correlation matrix; sample results based on Table 1.

| | I1 | I2 | I3 | I4 * | I5 | I6 | I7 | I8 * | I9 * | I10 * | I11 | I12 |
|---|---|---|---|---|---|---|---|---|---|---|---|---|
| I1 | 1.0000 | −0.0955 | 0.0354 | 0.0838 | −0.0127 | 0.1109 | 0.0129 | −0.0492 | 0.4208 | 0.3429 | 0.4265 | [ … ] |
| I2 | −0.0955 | 1.0000 | 0.0697 | −0.3778 | −0.2118 | −0.0464 | −0.0506 | 0.0533 | −0.0357 | −0.1504 | −0.0788 | [ … ] |
| I3 | 0.0354 | 0.0697 | 1.0000 | −0.0231 | −0.0205 | −0.0147 | −0.0167 | −0.0876 | 0.0235 | 0.0896 | 0.0111 | [ … ] |
| I4 * | 0.0838 | −0.3778 | −0.0231 | 1.0000 | 0.1147 | 0.0146 | 0.0097 | 0.0130 | 0.0466 | 0.0844 | −0.0051 | [ … ] |
| I5 | −0.0127 | −0.2118 | −0.0205 | 0.1147 | 1.0000 | −0.0443 | 0.0345 | −0.0245 | −0.0287 | 0.0512 | 0.0208 | [ … ] |
| I6 | 0.1109 | −0.0464 | −0.0147 | 0.0146 | −0.0443 | 1.0000 | −0.0062 | −0.0025 | 0.1103 | 0.1226 | 0.1061 | [ … ] |
| I7 | 0.0129 | −0.0506 | −0.0167 | 0.0097 | 0.0345 | −0.0062 | 1.0000 | −0.0803 | 0.0919 | −0.0316 | 0.0499 | [ … ] |
| I8 * | −0.0492 | 0.0533 | −0.0876 | 0.0130 | −0.0245 | −0.0025 | −0.0803 | 1.0000 | −0.0220 | −0.0073 | −0.0710 | [ … ] |
| I9 * | 0.4208 | −0.0357 | 0.0235 | 0.0466 | −0.0287 | 0.1103 | 0.0919 | −0.0220 | 1.0000 | 0.2737 | 0.4082 | [ … ] |
| I10 * | 0.3429 | −0.1504 | 0.0896 | 0.0844 | 0.0512 | 0.1226 | −0.0316 | −0.0073 | 0.2737 | 1.0000 | 0.2322 | [ … ] |
| I11 | 0.4265 | −0.0788 | 0.0111 | −0.0051 | 0.0208 | 0.1061 | 0.0499 | −0.0710 | 0.4082 | 0.2322 | 1.0000 | [ … ] |
| I12 | [ … ] | [ … ] | [ … ] | [ … ] | [ … ] | [ … ] | [ … ] | [ … ] | [ … ] | [ … ] | [ … ] | 1.0000 |

* obtained after adjusting the $h(x)$ function (Figure 1) with the values from associated vectors.

Explanations: * examples for I8, I9, I10, and other composite indicators transformed into single vectors.

- Is a particular word such as 'thing' present in the context? Detection;
- What type of thing is 'thing'? Classification;
- How could 'thing' be grouped or ungrouped? Segmentation

The scores retrieved by the machine are not important for the present article. However, based on the full set of values, the system can understand the connections between the vectors and decide what to do with the petition, as explained in Section 4.6. The real value

of such a system relies on the speed it can perform the triage for the incoming petitions and its accuracy, as will be presented below. Without it, the queue rate, the ratio of all petitions that are waiting for human observation, could scale up the capacity of the Integrated Technical Dispatch of Brasov city, seen here as a 'gatekeeper'. For example, one human annotator works 8 h per day, and a single annotation takes 5 min; then, a traditional system is capable of handling roughly 100 inputs per day, which is the system's capacity. One can perform the calculus and see that in the case of Brasov, the actual, traditional system exceeds the needs (1) [76].

$$12{,}935/251 \text{ working days per year in } 2022 \times 5 \text{ min} \approx 4 \text{ h } 20 \text{ min/day} \tag{1}$$

However, in case we are to expand the system to cover larger cities (for example, Bucharest, the Romanian capital, which is eight times larger than Brasov) or the entire country for specific central governmental agencies, the situation would be different. Moreover, during instances of a natural occurrence where unexpected surges may arise, humans typically lack the ability to promptly address the situation. Additionally, taking into account the time required to process information and respond to it, it is easy to envision the substantial benefits of such a system. Although queue rates can be unpredictable, the machine is undoubtedly capable of performing at a faster pace than humans and, with appropriate training, can achieve greater accuracy. Furthermore, machines do not rely on specific working hours, weekdays, or taking leaves.

In the picture above, in Figure 5, one can observe the color density, which shows the strength of the connections made by the system with words that are present in other petitions. In other words, the system is able to 'understand' the text in a more or less similar manner as humans do.

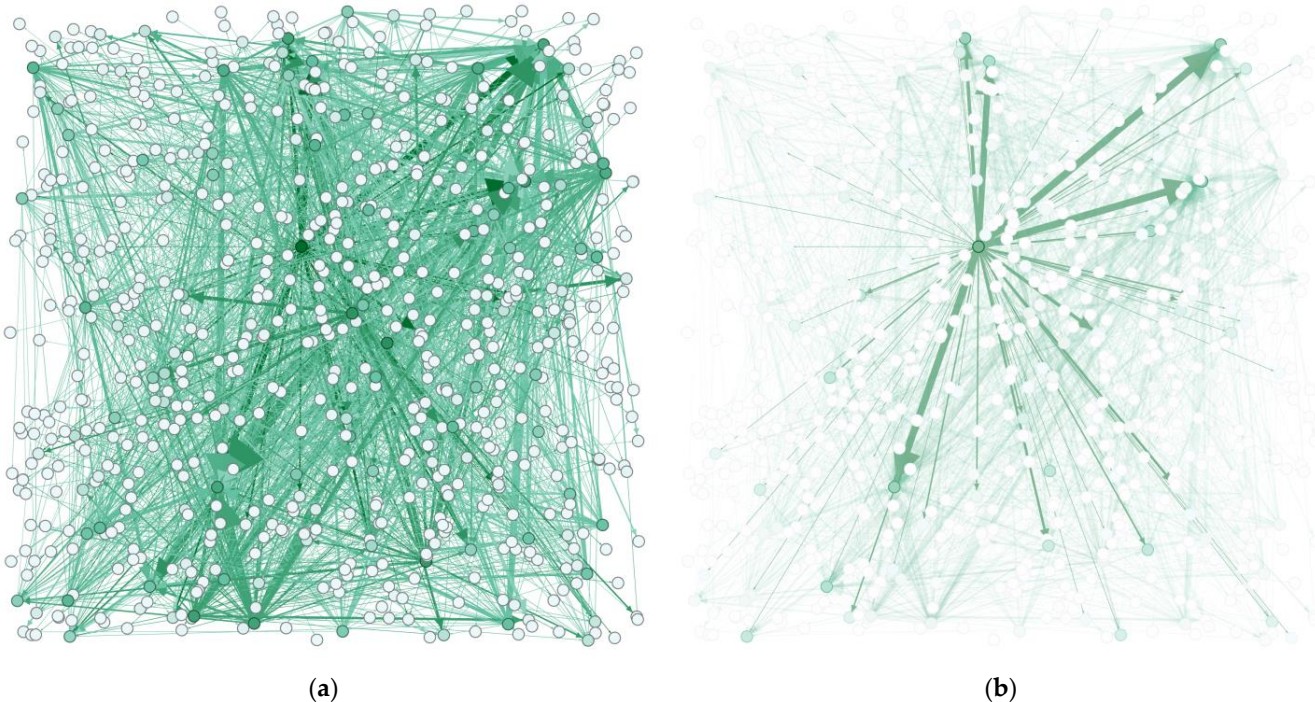

(**a**)         (**b**)

**Figure 5.** Analysis and visualization of keywords as they are found in the main corpus of 130 petitions. To provide readers with an understanding of the complexity, this 130-text sample (≈10% of the training set) is presented. It would have been impractical to use the full dataset for visual representation, as the excessive number of connections would have rendered the figure incomprehensible: (**a**) The complexity of the sample as seen by the machine; (**b**) An example of one-word (in this case it was chosen 'transportul'–Romanian for 'transportation') connections with context in other petitions.

After the strengths have been computed, the machine typically has the ability to assess the petitions, albeit with some degree of inaccuracy, as shown in Figure 6 below, and decide the action. For example, if one complaint is addressing water linkage in one neighborhood of the city, the petition might contain words such as 'water', 'pipe', 'road', and others in this category but is unlikely to have 'thefts', 'wild animals', 'traffic lights', and so on. When these instances appear, however, the system will forward the text to a human operator observing his/her behavior and (re)adjusting the model. Moreover, human operators can help the machine with these adjustments for a better future prediction.

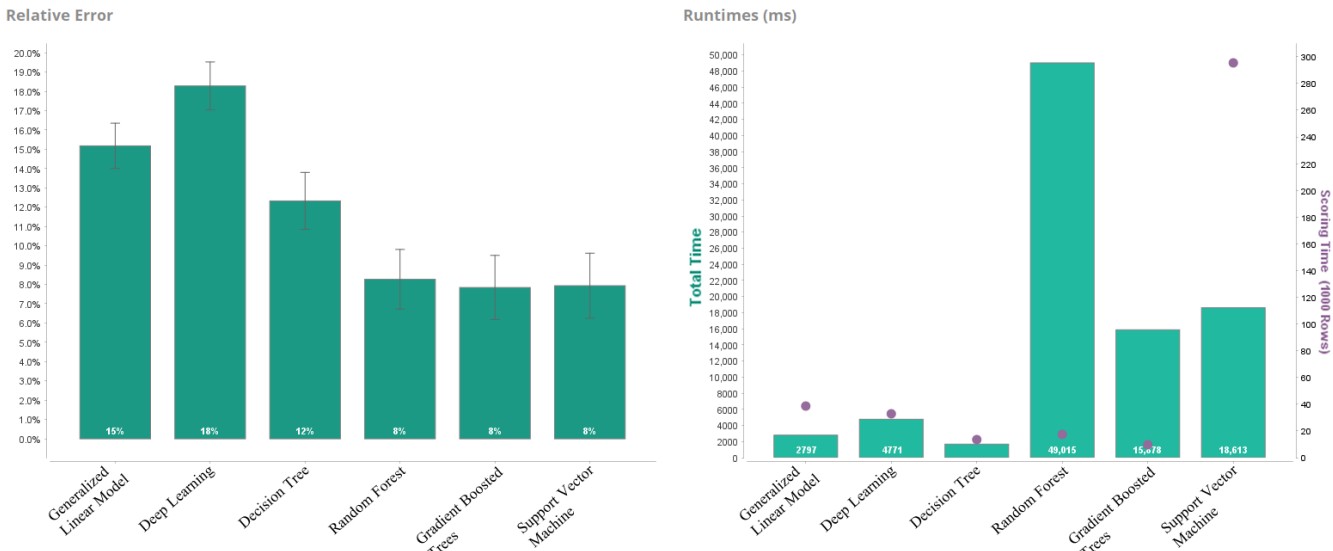

**Figure 6.** Prediction models used experimentally for this article.

The chart depicted in Figure 6 shows the results of testing six distinct prediction models. The Gradient Boosted Tree model proved to be the most precise, with a relative error of 7.84%, but also had a moderate efficiency, taking approximately 16 s to process the text and determine its course of action. In comparison, other models, such as the Decision Tree, were faster, taking only 2 s, but had a higher error rate of 12.32%.

The author would like to clarify that the intention was not to measure the error rate of the city hall experts, which may have been lower than any of the models tested in the study. However, the results did support the hypothesis that machine speed could be an advantage, and with adequate training, the accuracy of these models could also improve.

## 6. Discussion

The advancement of technology has enabled AI to make remarkable strides in managing critical aspects of 'compare and comply' functions. In their activities, public administration officers scrutinize lots of data, mostly legislation and internal norms, for being able to avoid unforeseen legal complications. This is still a rather difficult problem to be solved by machines since specific concepts can be formulated in different ways. However, the system is not trying to replace humans but to help them perform better; therefore, the critics that are targeting AI systems mistakes, known as 'adversarial examples', are not to be seen as bugs but as features [77]. Nevertheless, the role of automation is to make the tasks easier by allowing software to scan legislative documents, understand the meaning, compare it with the citizen demand, and determine which documents are to be referred to in the answer, resulting in significant time and effort savings.

The methodology employed, as described in the article, could potentially be adapted to the sentiment analysis problem associated with e-petitions, contingent upon access to a Romanian-language lexicon that includes positive and negative terms (teams of experts in this field are working right now on building this [78,79]). Identification of the most extreme

positive and negative terms will enable the classification of 'obvious' entries, with the self-supervised approach subsequently handling the remaining entries by cross-referencing them against these extreme examples.

Apart from the e-petition issue referenced, the techniques detailed in the article could prove applicable to a diverse range of other-related challenges. Specifically, it would entail classifying the simplest, most extreme text entries as positive or negative, utilizing the word lexicon, and subsequently using these outputs as labels for machine learning classifiers. The remaining text entries could then be processed through the classifier to obtain positive, negative, neutral, or uncertain classifications [80,81].

The study results indicate that automation can lead to significant time and effort savings by enabling software to scan legislative documents, comprehend their significance, and contrast them with citizen demands. Although machines cannot entirely replace humans, they can substantially augment their abilities and efficiency in handling intricate tasks.

### 6.1. Limitation

The efficiency of the system was mentioned. Of course, this is a debatable issue since the machines are not a panacea. The system is far from perfect. Nonetheless, if there is a rare occurrence of a false positive or false negative, as seen in Figure 4, the system can request additional information from the citizen or escalate the issue to a human operator. It is crucial to involve humans in AI systems to ensure accountability and accuracy; in other words, caution is required.

Moreover, in a live environment, the system may potentially misbehave; biases might pop up, and that could jeopardize the output resulting in court cases. In order to avoid this, it is important to identify and address potential biases to ensure fairness and ethical use of the system. Bias can arise from a variety of sources, such as imbalanced training data, algorithmic limitations, etc. Failing to address these biases can result in discrimination against certain groups of people or in inaccurate predictions resulting in bad outputs, which can have serious consequences, including, as mentioned, legal action. To mitigate the risk of bias, it is important to establish best practices for data labeling by experts and preprocessing by administrators, together with ongoing monitoring and evaluation of the system's performance. This can include techniques such as data augmentation, model interpretability, and fairness metrics, as well as involving diverse actors in the design and implementation of the system.

### 6.2. Future Work

In the case of petition analysis, the system should be trained on a large corpus of text data, which should be labeled with relevant metadata such as issue type, urgency, sentiment analysis, and many others. By analyzing the patterns in the data, the system can learn to identify common themes and topics and make connections between related words and phrases. This allows the system to accurately classify new petitions and prioritize them based on their level of urgency and importance.

Moreover, in an e-petition platform enhanced with AI, GPT-4 API (Application Programming Interface) could be integrated as an intelligent chatbot to provide personalized and efficient customer support. Newly released GPT-4, as a large language model (Generative Pre-trained Transformer), has the ability to understand and respond to natural language queries, making it an ideal candidate for handling user inquiries in real time.

As an idea for future work, the integration of GPT-4 in the petition system can be further enhanced by incorporating advanced machine learning techniques. This can improve the accuracy and relevance of GPT-4's responses by enabling it to learn from user feedback and adapt to changing user needs.

Additionally, the author is willing to mention that at the present stage, the system developed together with the present article has no graphical user interface, being mostly an algorithmic approach to an AI problem. However, the present paper does not consist in

promoting this one in particular but in developing apps capable of helping public servants, institutions, and, in the end, the citizens.

The application of transfer learning and self-supervised learning techniques would prove especially advantageous in the implementation of such a system, which could then be utilized by other public administration entities, including museums and other institutions that serve citizens directly. With consistent input data, it would be feasible to modify the output layer to gain a deeper understanding of citizens' needs without the need to start anew.

While the financial aspect may not be immediately evident in public management, the use of advanced technologies has the potential to generate tangible benefits, such as increased citizen trust and engagement.

### 6.3. Theoretical, Practical, and Policy Implications

Theoretical implications suggest that the model developed in this study could potentially be adapted to solve the sentiment analysis problem associated with e-petitions. Moreover, practical implications reveal that AI has the capability to assist public administration officers in managing large volumes of data, saving significant time and effort. By scanning legislative documents, comprehending their meaning, comparing them with citizen demands, and determining which documents to reference in the response, AI can alleviate the workload of public servants by swiftly and accurately processing vast amounts of text 24/7.

Overall, this study demonstrates that AI can effectively assist public administration officers in managing large amounts of data while also identifying potential biases and ensuring ethical use of the system. Furthermore, AI has the potential to generate tangible benefits, increasing citizen trust and engagement, and can be employed in other public administration entities as well.

### 7. Conclusions

The author conducted this experiment in order to explore the computations involved in the context of learning how to operate text-based inputs. The findings suggest that these models can, theoretically, be implemented and empirically execute a range of actions depending on the model's capacity and noise in the dataset, seen here as blurry text sequences inside petitions. Furthermore, it was demonstrated that the AI models could ease the workload of public servants by computing large amounts of text with high speed and accuracy seven days a week 24 h per day. While the experiment was centered on linear functions, with relatively few layers and indicators, seen here as vectors, the methodology can be extended to many other learning problems involving richer function classes. For instance, it can be applied to a network that performs non-linear feature computation in its initial layers.

Additionally, this experimental approach can be used to study larger-scale examples of contextual learning, such as language models, and determine whether their behaviors can be explained by interpretable learning algorithms. Although there is still much work to be done, the results provide initial evidence that what today is seen as an online but asynchronous way of dealing with citizens' requests, in the future, may not be as difficult as it seems and can be put in practice using standard machine learning tools. Furthermore, the solutions provided by the artificial intelligence tools will help in creating better communication with the public administration and finding better solutions to citizens' problems. Implementing NLP techniques in public administration processes is just one of the first steps in the e-government 3.0 era.

**Funding:** This research received no external funding.

**Institutional Review Board Statement:** Not applicable.

**Informed Consent Statement:** Not applicable.

**Data Availability Statement:** Not applicable.

**Acknowledgments:** The author would like to express sincere gratitude to the staff of the Brasov City Hall Computer Department for their invaluable assistance in providing the necessary information and insightful discussions on the software applications. Their support and guidance provided the author with a strong foundation for the development of this article.

**Conflicts of Interest:** The author declares no conflict of interest.

## Appendix A

**Table A1.** Petitions by state and category *.

| ID | Item | Active | Resolved | Total |
|----|------|--------|----------|-------|
| 1 | Unauthorized display/trade | - | 34 | 34 |
| 2 | Road improvements | 244 | 1250 | 1494 |
| 3 | Animals in public domain | 2 | 58 | 60 |
| 4 | Damage to utility networks | 312 | 1653 | 1965 |
| 5 | Requests for information | 9 | 2520 | 2529 |
| 6 | Unauthorized construction/works | - | 160 | 160 |
| 7 | Waste disposal | 2 | 107 | 109 |
| 8 | Destruction of public domain | 6 | 58 | 64 |
| 9 | Fountain | - | 6 | 6 |
| 10 | Public lighting | 3 | 851 | 854 |
| 11 | Investments | 10 | 5 | 15 |
| 12 | Road markings | - | 18 | 18 |
| 13 | Illegal parking | 5 | 882 | 887 |
| 14 | Public/residential parking | 5 | 159 | 164 |
| 15 | Free passage permit | - | 16 | 16 |
| 16 | Environmental issues | - | 77 | 77 |
| 17 | Sanitation | 6 | 1443 | 1449 |
| 18 | Road signs | 2 | 970 | 972 |
| 19 | Electronic services/Web portal | 12 | 18 | 30 |
| 20 | Administrative Service Complaints | 9 | 4 | 13 |
| 21 | Emergency situations | - | 28 | 28 |
| 22 | Public transport | 43 | 142 | 185 |
| 23 | Taxi transport | - | 4 | 4 |
| 24 | Public disturbance | 2 | 356 | 358 |
| 25 | Abandoned vehicle | 1 | 318 | 319 |
| 26 | Zero plastic in green areas | - | 8 | 8 |
| 27 | Green areas/urban furniture | 12 | 1790 | 1802 |
| | Total | 685 ** | 12,935 | 13,620 |

* Integrated Technical Dispatch–general activity report for the period 1 January 2022–31 December 2022; ** 685 requests were registered in late December. They were mostly tackling issues that stemmed from Brasov's high-altitude location, which causes massive snowfalls during the winter.

**Table A2.** Petitions by origin *.

| ID | Item | Total |
|---|---|---|
| 1 | Email | 968 |
| 2 | Smartphone | 8074 |
| 3 | Instant message | 2 |
| 4 | Web platform | 1463 |
| 5 | Phone | 3113 |
| | Total | 13,620 |

* Integrated Technical Dispatch; general activity report for the period 1 January 2022–31 December 2022.

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
