# Peer review of "E-Government 3.0: An AI Model to Use for Enhanced Local Democracies"

_sustainability, doi:10.3390/su15129572_

Round 1

Reviewer 1 Report

Dear author, 

your project where you aim to analyse e-government 3.0, which is based on the principles of 2.0 but refers to the use of new emerging technologies, is really interesting and the presentation of a new model can also make an important contribution to the literature.

However, I think the article needs to be revised before possible publication in this journal.

Some important suggestions:

1. The article is not properly referenced, there are many articles in the literature that address this issue. Therefore, I recommend reviewing the introduction and literature review section in this respect.

2. English needs to be improved

3. Check the citations according to the journal standard

4. I really have difficulty reading and understanding Fig. 1 (Figure 1. Analysis and visualisation of keywords (in Romanian language) as they are found in the 239

subject lines of petitions - first classification task) and the following comment.

5. Fig. 3 and Fig. 6 are also very incomprehensible (which need to be commented on better, using the literature).

I really appreciated the discussion section, I think the inclusion of future work and limitations is really good work.

In conclusion, it is a good article, but revisions are needed.

Best regards

Author Response

Dear author, 

your project where you aim to analyse e-government 3.0, which is based on the principles of 2.0 but refers to the use of new emerging technologies, is really interesting and the presentation of a new model can also make an important contribution to the literature.

The author would like to express gratitude to the reviewer for their thoughtful and encouraging feedback. Their kind words are greatly appreciated and have served as a valuable source of motivation for the author.

However, I think the article needs to be revised before possible publication in this journal.

Some important suggestions:

  1. The article is not properly referenced, there are many articles in the literature that address this issue. Therefore, I recommend reviewing the introduction and literature review section in this respect.

The author tried exploring the vast amount of literature, but as the reviewer noted, there exists a substantial body of literature on the topic of e-gov. While the author has included additional information, he would appreciate the reviewer's assistance in identifying relevant titles that could enhance the article. Any recommendations would be greatly appreciated. Thank you in advance for your kind help!

  1. English needs to be improved

The author has carefully reviewed and edited the manuscript, ensuring that any errors have been addressed. However, should the reviewer come across any further mistakes, the author kindly requests that they be pointed out for prompt correction. Thank you for your valuable feedback.

  1. Check the citations according to the journal standard

The author has reviewed the manuscript and made necessary adjustments to ensure its accuracy. Thank you for your time and effort in providing feedback on this work.

  1. I really have difficulty reading and understanding Fig. 1 (Figure 1. Analysis and visualisation of keywords (in Romanian language) as they are found in the 239

subject lines of petitions - first classification task) and the following comment.

The author has made modifications to Figure 1 in accordance with the reviewer's suggestions. However, it is still challenging to comprehend due to the numerous words and limited space available to display them. The most crucial aspect of the image is for the reader to grasp and comprehend the connections established by the machines, similar to how young children (<2 years) connect words based on who is speaking and to whom they are addressing them. While this explanation in particular may not be suitable from a scientific perspective and lacks references, the concept is comparable.

  1. Fig. 3 and Fig. 6 are also very incomprehensible (which need to be commented on better, using the literature).

The author has made modifications to both figures based on the reviewer's recommendations. It is presumed that the reviewer's concern with Figure 6 was related to the small font size, which was unavoidable due to it being a simple print screen of the program's visual interface. However, the author has since edited the image using an editor and enlarged the font size to enhance its clarity. Thank you for your valuable feedback. As author, it can be challenging to identify such issues since my focus was primarily on writing and while doing so, the content appears clear in my mind. Thank you for your suggestion.

I really appreciated the discussion section, I think the inclusion of future work and limitations is really good work.

Thank you very much for your encouraging words!

In conclusion, it is a good article, but revisions are needed.

Again, thank you very much! 

Best regards

Reviewer 2 Report

* In the paper form: Wellorganised, just some grammarand spelling to revise.

* The content :

1. Abstract: Author kindly follow this methodology to improve the abstract  (background; objective, method used, findings, original value, limitations and practical users).

2. Literature review: it is not enaugh. add more 2 to three paragraphs of the relevent studies related to yours.

3. Findings (materials): 

* Sample :Romanian cities – Brasov- : (it'll well better join comparaision other cites)

* Time series: One year (01.01.2022-31.12.2022) : More the periode is long, more the  variables results are statistical significants.

* Table1 : ID - I12 […] ; I13 […] : is  not clair !!

* Figure 1. Analysis and visualization of keywords (in Romanian language): is not clair and send us the input data of this figure1

* Figure 2. Generic model of predicting the importance (set as vector) : send us the input data of this figure2

5. Results :

 * Table 2. Correlation matrix – sample results based on Table 1 : Kindly provide the software used and send the input data (Table 2).

* Figure 5 (a&b) : are note clair).

6. Discussion : to improve more and more. ( discuss more the IA revolution technology apport to this study -original value-implications)

7. Conclusion : reorganise it in two pragraphes.

Good luck

Author Response

* In the paper form: Wellorganised, just some grammarand spelling to revise.

The author would like to express their sincere appreciation for the reviewer's kind and supportive feedback. Thank you very much for your thoughtful comments, which have served as a source of motivation for the author.

* The content :

  1. Abstract: Author kindly follow this methodology to improve the abstract  (background; objective, method used, findings, original value, limitations and practical users).

The author has made changes to the manuscript abstract based on the reviewer's feedback. Thank you for your valuable input, which has helped improve the quality of the work.

  1. Literature review: it is not enaugh. add more 2 to three paragraphs of the relevent studies related to yours.

The author has included an additional paragraph in the introduction, as recommended by the reviewer. However, since the literature on this topic is extensive, the author would appreciate the reviewer's assistance in identifying relevant studies that could further enhance the article. Any recommended references would be greatly appreciated. Thank you in advance for your valuable help!

  1. Findings (materials): 

* Sample :Romanian cities – Brasov- : (it'll well better join comparaision other cites)

The author would like to express gratitude for the reviewer's comment. It should be noted that the study is not of a sociological nature. The NLP (Natural Language Processing) technique utilized in this study does not behave differently when the dataset is extended to a different city. All that is required is text organized as words in a dictionary, as well as sentences written in natural language to comprehend the connections between words. This technique can be applied to any type of text, including customer reviews from any Romanian online product delivery portals (such as www.emag.ro), and will produce consistent results.

* Time series: One year (01.01.2022-31.12.2022) : More the periode is long, more the  variables results are statistical significants.

The author would like to express appreciation for the reviewer's comment. It is true that big data can be beneficial. However, for the purpose of this article, the author believes that one full year of data is sufficient. The duration of the study does not affect the results since it is not a sociological study. Rather, its aim is to demonstrate the speed at which text analysis can be performed as compared to human analysis. The time period mentioned in the article is only a reference for the time commitment required and not intended to affect the data collection process. One month or even one week of Amazon customer reviews would likely yield similar outcomes.

* Table1 : ID - I12 […] ; I13 […] : is  not clair !!

The meaning of […] is to demonstrate to the reader that there are numerous other indicators (a total of 118 organized in 47 different categories of analysis) that are following, but due to limited space, it is not possible to present detailed information. Furthermore, the mathematical calculations involved are quite complex, and the paper is not intended to be technical but rather to inform policy makers. Nevertheless, an explanation has been included to address this issue. Thank you for your suggestion!

* Figure 1. Analysis and visualization of keywords (in Romanian language): is not clair and send us the input data of this figure1

Attached to my answer the reviewer can see the data… however, please note that the data attached to my response is not intended for public use, and should not be used or published outside of the scope of this study. Thank you for understanding.

* Figure 2. Generic model of predicting the importance (set as vector) : send us the input data of this figure2

Attached to my answer the reviewer can see the data… however, please note that the data attached to my response is not intended for public use, and should not be used or published outside of the scope of this study. Moreover, two data augmentation techniques were deployed here in order to keep it safe from being used outside the original platform but still to be confronted with other software to validate the outputs. Two of the articles used as a reference for augmenting data is to be found here: https://arxiv.org/abs/1311.2901, and here: https://arxiv.org/abs/2002.05709. Since, however, the article is not a technical one, all those mathematical and computational models simply don’t fit here. The author hope that the reviewer sees the complexity and agrees that the article is not a technical one but meant to address readers such as policy makers in order to make them understand that emerging technologies are here to help public administration processes.

  1. Results :

 * Table 2. Correlation matrix – sample results based on Table 1 : Kindly provide the software used and send the input data (Table 2).

The software used is to be found here: https://github.com/google-research/bert, but adjusted with the data used for this article (also attached a table with relevant data extracted from the huggingface hub platform - https://huggingface.co/, providing the value of BERT app). However, the training set used for the purpose of this article is of course different. Also, for the purpose of obtaining the outputs, the author was using Google colab - https://colab.research.google.com/. On the platform one can see a sample data with the results provided – if the settings (also the steps) are as proposed by the platform.

For tabular data obtained after processing the author was using a slightly modified version of this software: https://github.com/dreamquark-ai/tabnet. They are all in open-source format.

Additionally, while submitting this article, the author is simultaneously working on a more technically oriented piece to be sent to specialized journals such as MDPI Systems, Mathematics, or Applied Sciences, and plans to publish the software used to build it on GitHub in the hopes of garnering interest from interested parties. It should be noted that the status of these efforts may be subject to change due to financial constraints.  

Attached to my answer the reviewer can see the data… however, please note that the data attached to my response is not intended for public use, and should not be used or published outside of the scope of this study. The reviewer can refer to the tables to gain access to a comprehensive breakdown of the training data set, the dictionary employed for data analysis, the augmented weights (computed using the methods mentioned earlier), as well as a list of keywords identified in the subject lines of the e-petitions.

* Figure 5 (a&b) : are note clair).

The figures serve to provide an overview of the intricacies involved in a language model's functioning, which is similar to that of a 2-year-old child's attempt at comprehending natural languages. Although this explanation may not be scientific, machines essentially operate in much the same way. They analyze words and their contexts by drawing connections between them (for instance, it is improbable for words like 'giraffe' and 'swimming pool' or 'bird cage' to appear in the same sentence, and therefore the system will assign low weights to such combinations. Conversely, words like 'lion' and 'savannas' are more likely to be correlated with high weights, as demonstrated in one of the accompanying data sets) but also in the paper in Table 1.

  1. Discussion : to improve more and more. ( discuss more the IA revolution technology apport to this study -original value-implications)

As recommended by the reviewer, the Discussions section has been enriched.

  1. Conclusion : reorganise it in two pragraphes.

Corrected as suggested by the reviewer.

Good luck

Thank you very much for your kind advices.

Reviewer 3 Report

Unfortunately I found this paper very hard to follow mostly because I had hard time understanding what the paper tried to achieve. The paper did not appear to follow the normal research process of 1) Raising research questions or hypotheses 2) Finding logical support (references) for deriving  these questions or hypotheses 3) Establishing proper research design stating how the analysis was carried out, and 4) Drawing conclusions and discussing implications.

It seems that authors tried to explain post-facto a result of running AI-based program which is supposed to help increase the petition processing efficiency at local government. However,I could not find any part of the paper dealing with the clear research purpose, and how the end result relates to intended research purpose. It simply shows how the authors tried a program and solved a problem (increasing process efficiency) without telling us much about the research issues (i.e., what is the academic or practical implications (contributions) of the paper. 

I suggest authors submit this paper to a technically oriented journal which has a focus on improvement of a technology based system such as AI-based program. Another outlet would be to try to submit it as a short communications for a technology oriented journals.  

Author Response

Dear reviewer, I would like to express my sincere appreciation for your valuable insights and the effort you put into reviewing my paper. Your comments have been incredibly helpful in improving the overall quality of the article.

I would like to address some concerns you raised regarding the technical aspect of the paper. Although I have a technical background, my academic career has been dedicated to translating the complexities of the digital world into easily digestible documents such as policies and strategies for public administration, in addition to teaching university courses. As the author of this article, I was mindful not to make it too technical, assuming that the readers consist of experts, professionals in social sciences, public administration, economics and most probably policy makers. However, while submitting this article, the author is simultaneously working on a more technically oriented piece to be sent to specialized journals such as MDPI Systems, Mathematics, or Applied Sciences, and plans to publish the software used to build it on GitHub in the hopes of garnering interest from interested parties. It should be noted that the status of these efforts may be subject to change due to financial constraints.

Furthermore, I believe that the Sustainability journal is widely used as a scientific reference by many policy makers worldwide, and therefore the paper is not only relevant to the subject matter but also suitable for the reader's profile.

Once again, I would like to thank you for your thoughtful comments, and I look forward to incorporating your suggestions into the revised version of the article.

Issue 1 (I1). Unfortunately I found this paper very hard to follow mostly because I had hard time understanding what the paper tried to achieve. The paper did not appear to follow the normal research process of 1) Raising research questions or hypotheses 2) Finding logical support (references) for deriving  these questions or hypotheses 3) Establishing proper research design stating how the analysis was carried out, and 4) Drawing conclusions and discussing implications.

Answer 1 (A1). The paper follows a well-defined research process, starting with an introduction that provides context for the research and a brief summary of each section and subsection. The hypotheses are also addressed in the introduction, as seen on line 90 and onwards. Scientific support is provided not only in the introductory section but also in the literature review. However, due to the extensive number of articles available in the e-gov research field, the author kindly requests the reviewer's assistance in identifying relevant articles on the topic.

The research design is presented in section 4, building on the methods outlined in section 3. The conclusions and discussions follow the results section. The author respectfully requests the reviewer's help in resolving any issues that may arise in these sections. Thank you for your time and consideration.

I2. It seems that authors tried to explain post-facto a result of running AI-based program which is supposed to help increase the petition processing efficiency at local government. However,I could not find any part of the paper dealing with the clear research purpose, and how the end result relates to intended research purpose. It simply shows how the authors tried a program and solved a problem (increasing process efficiency) without telling us much about the research issues (i.e., what is the academic or practical implications (contributions) of the paper. 

A2. Thank you for your valuable comment. I would like to clarify that the results, as outlined in the Material and Methods section, were obtained through an experiment with an innovative model, or instrument if the reviewer is more comfortable with this concept, that aims to help public administrations keep up with the technologies used by citizens and engage with them more effectively. While the system/ platform is not currently deployed in the real world, it serves as a model, or framework if this word fits better, for future development. It is worth mentioning that this model differs from typical customer reviews platforms found on major online stores, as it focuses on a wider range of topics and is subject to various legal norms and regulations. On the other hand, sentiment analysis techniques (references added in the paper) applied to consumer reviews can provide valuable insights into how to improve overall service or products.

I have rephrased the paragraphs starting from the abstract to better address the research purpose and hope that the reviewer will find them clearer and more comprehensible. Once again, thank you for your helpful feedback.

I3. I suggest authors submit this paper to a technically oriented journal which has a focus on improvement of a technology based system such as AI-based program. Another outlet would be to try to submit it as a short communications for a technology oriented journals.  

A3. As previously mentioned, a technical article explaining the engine of the platform will follow shortly. At present, the author is focusing on fine-tuning the software. The final product will likely be made publicly available on GitHub as open-source software that can be adjusted and utilized by anyone seeking to improve it or deploy it as a software-as-a-service (SaaS). However, it should be noted that this is subject to change.

To provide some context and a better understanding of the purpose of this paper, it's worth mentioning that e-governance has evolved over time. It started with publishing information on the web (first stage), followed by sending and receiving emails by the municipality (second stage), building bidirectional platforms (third stage), allowing e-payments (fourth stage), and enabling citizens to participate in governance through online means (social media or other platforms like Zoom, Webex, and so on – fifth stage). With the increasing popularity of AI, it is now seen as a new stage (sixth) in e-governance development. It's important to note that this paragraph, as well as the article itself, is not meant to be a boring e-governance lesson but rather an attempt to place the article in the context of e-governance development over time. The author provides scientific context in both the sphere of e-governance/public administration and computer science within the article, without delving too deeply into either, as the article targets the intersection of these sciences.

The author would like to express his sincere appreciation for the reviewer's time and effort spent on providing feedback for this paper. He hopes that his responses have adequately addressed the concerns raised by the reviewer and improved the quality of the paper. Thank you again for your valuable contributions.

Reviewer 4 Report

The paper brings into attention a topic of interest regarding the use of artificial intelligence in public administration. The author has performed a case study on e-petition systems.

It is not clear which are the hypotheses of this paper. Usually, hypotheses are used in quantitative studies, being tested in the data analysis phase. The author has employed a qualitative method, case study, and the hypothesis mentioned in  section 4 is not similar with those mentioned in the introduction.

I recommend to present some theoretical coordinates regarding the method employed in this paper: case study. 

The conclusions might be extended, by presenting some managerial implications in a more specific manner.

Author Response

I would like firstly to express my thanks to the reviewer for his kind words and valuable suggestion.

The paper brings into attention a topic of interest regarding the use of artificial intelligence in public administration. The author has performed a case study on e-petition systems.

It is not clear which are the hypotheses of this paper. Usually, hypotheses are used in quantitative studies, being tested in the data analysis phase. The author has employed a qualitative method, case study, and the hypothesis mentioned in  section 4 is not similar with those mentioned in the introduction.

The author made adjustment both on the hypotheses section (Introduction) as well as in the rest of the text.

I recommend to present some theoretical coordinates regarding the method employed in this paper: case study.

Also, here the author performed a couple of adjustments hoping it meets the reviewer rigor.

The conclusions might be extended, by presenting some managerial implications in a more specific manner.

There were added few more information in the Discussion section in regard to this topic. The author hopes that the reviewer considers the new input as valuable as it would have been in the Conclusion section.

Reviewer 5 Report

The author presents a study on the potential of e-government 3.0 and AI in transforming public service delivery and governance, with a focus on enhancing citizen participation and improving administrative systems. The paper proposes an AI-based model that utilizes government data from Romania. Although the benefits of employing AI technologies in providing services to citizens are apparent, the paper lacks rigor in its analysis, making it unsuitable for publication in this journal.

To improve the rigor of the analysis, the paper could have employed various evaluation methods, such as cross-validation, testing on independent datasets, and measuring performance metrics like accuracy, precision, recall, and F1-score. Additionally, the paper could have considered factors such as the quality and size of the training data, model complexity, and potential biases in the data. These measures would have ensured the quality and usefulness of the proposed model. Alternatively, the paper could have presented a conceptual model, which could have been evaluated based on expert review, peer review, validation, simulation, and sensitivity analysis. While the paper provides some background on AI concepts, the proposed model does not fit into either a conceptual or algorithmic framework. Moreover, the paper does not present in detail the evaluation methods used to assess the author's approach.

In sum, to improve the paper's quality, the author should consider including a more rigorous analysis and evaluation of the proposed model. Additionally, the author could present a conceptual model that supports the study's objectives and evaluate it using appropriate methods.

Author Response

The author presents a study on the potential of e-government 3.0 and AI in transforming public service delivery and governance, with a focus on enhancing citizen participation and improving administrative systems. The paper proposes an AI-based model that utilizes government data from Romania. Although the benefits of employing AI technologies in providing services to citizens are apparent, the paper lacks rigor in its analysis, making it unsuitable for publication in this journal.

After a thorough evaluation of the paper and receiving five additional reviews, the author has significantly improved the content. The purpose of these revisions is to ensure that the paper meets the rigorous standards of the reviewers. Notably, the updated paper is more appropriate for readers interested in sustainability, particularly non-technical readers such as policymakers who are keen on the rapid development and deployment of technology in the public sector. With AI currently generating considerable interest and traction, it is expected to be widely adopted and used by public servants and citizens. Despite some existing barriers, AI has enormous potential to benefit both groups.

To improve the rigor of the analysis, the paper could have employed various evaluation methods, such as cross-validation, testing on independent datasets, and measuring performance metrics like accuracy, precision, recall, and F1-score. Additionally, the paper could have considered factors such as the quality and size of the training data, model complexity, and potential biases in the data. These measures would have ensured the quality and usefulness of the proposed model. Alternatively, the paper could have presented a conceptual model, which could have been evaluated based on expert review, peer review, validation, simulation, and sensitivity analysis. While the paper provides some background on AI concepts, the proposed model does not fit into either a conceptual or algorithmic framework. Moreover, the paper does not present in detail the evaluation methods used to assess the author's approach.

Thank you for your valuable suggestion. To conduct our analyses, we employed regression models utilizing various machine learning techniques, including Generalized Linear Model, Deep Learning, Decision Tree, Random Forest, Gradient Boosted Trees, and Support Vector Machines, as evidenced in Figure 6. However, it is important to note that this article is not intended to be a technical or mathematical piece, nor is it focused on sociological studies. Rather, its aim is to present a novel approach to addressing administrative issues, specifically e-petitions.

While the techniques recommended by the reviewer are undoubtedly impressive and well-suited for sociological studies, our paper centers on harnessing computing power to assist civil servants. This approach could prove highly effective for analyzing customer reviews on an online selling platform, for example, enabling the machine to interpret customer inputs and providing platform administrators with insight into customer attitudes (positive, negative, or neutral) towards their products to improve quality.

It is worth mentioning, however, that our conceptual model is not without limitations. As mentioned in the paper, the system requires thorough testing before release into the real world and ongoing supervision to continually refine and enhance accuracy. We have revised the paper to make this information more transparent and comprehensible to the reader. Once again, thank you for your valuable feedback.

In sum, to improve the paper's quality, the author should consider including a more rigorous analysis and evaluation of the proposed model. Additionally, the author could present a conceptual model that supports the study's objectives and evaluate it using appropriate methods.

Thank you for your feedback. As the author, I appreciate your suggestions for improving the quality of the paper. According to your recommendations I adjusted the analysis and evaluation of the proposed model. Furthermore, a clear and more concise explanations of the model was provided. Your feedback has been valuable in helping me to enhance the paper's quality, and I am grateful for your contribution to my work.

Please receive my kindest regards!

Reviewer 6 Report

Dear Authors,

The paper focuses on the evolution of e-government from the first generation to the third generation. The author notes that the term 'e-government 2.0' is becoming less common as the focus shifts towards broader digital transformation initiatives that may include emerging technologies such as artificial intelligence, blockchain, virtual reality, and augmented reality.

The paper analyzes the challenges faced by municipalities in responding to citizen petitions, which are a core application of local democracies. The author presents an example of an e-petitions system currently in use and proposes an AI model that can deal faster and more accurately with the increased number of inputs. The author intends to promote the adoption of AI by municipalities that are still reluctant to implement it in their operations.

After having reviewed your manuscript, we have identified several areas for improvement that will help enhance the clarity and organization of your paper. Please find below a summary of my comments on each section:

Introduction:

1.     Provide more specific research questions or hypotheses.

2.     Organize the introduction more clearly.

3.     Use more reader-friendly language.

Literature Review:

4.     Improve organization and coherence.

5.     Include more recent research on the use of AI in e-petitioning and governance processes.

6.     Provide more concrete examples of the use of AI in e-petitioning.

Materials and Methods:

7.     Provide more detailed descriptions of the methods used to clean and process the data set.

8.     Discuss potential limitations of the study, particularly on the validity of the data set.

The AI Model Proposed:

9.     Add background information on the use of AI in the legal field.

10.  Simplify language and provide more explanations.

11.  Include a broader discussion on ethical considerations of using AI for petition classification.

Results:

12.  Provide more information about the results obtained.

13.  Discuss limitations and suggest areas for future research.

Discussion:

14.  Explore the ethical implications of biased decision-making.

15.  Provide more concrete examples of the use of AI in real-world applications.

16.  Provide a more detailed exploration of limitations of AI systems in public administration.

Conclusion:

17.  Expand upon implications for practice and discuss how findings could be applied in real-world settings.

I hope these comments will be helpful to you in revising your manuscript.

Please receive our kindest regards,

Author Response

Dear Authors,

The paper focuses on the evolution of e-government from the first generation to the third generation. The author notes that the term 'e-government 2.0' is becoming less common as the focus shifts towards broader digital transformation initiatives that may include emerging technologies such as artificial intelligence, blockchain, virtual reality, and augmented reality.

The paper analyzes the challenges faced by municipalities in responding to citizen petitions, which are a core application of local democracies. The author presents an example of an e-petitions system currently in use and proposes an AI model that can deal faster and more accurately with the increased number of inputs. The author intends to promote the adoption of AI by municipalities that are still reluctant to implement it in their operations.

After having reviewed your manuscript, we have identified several areas for improvement that will help enhance the clarity and organization of your paper. Please find below a summary of my comments on each section:

Firstly, allow me to express gratitude to the reviewer for their thoughtful and encouraging feedback.

Introduction:

  1. Provide more specific research questions or hypotheses.

The author has carefully reviewed the hypotheses paragraph and added more info.

  1. Organize the introduction more clearly.

One more paragraph entered the Introduction section. This will provide more context in the urban technologies sphere.

  1. Use more reader-friendly language.

The author has reviewed the manuscript and made necessary adjustments to ensure a clear understanding.

Literature Review:

  1. Improve organization and coherence.

The author has reviewed the manuscript and made necessary adjustments to ensure a clear understanding.

  1. Include more recent research on the use of AI in e-petitioning and governance processes.

The author has included few more studies on the topic, as recommended by the reviewer. However, since the literature on this topic is extensive, the author would appreciate the reviewer's assistance in identifying relevant studies that could further enhance the article. Any recommended references would be greatly appreciated. Thank you in advance for your valuable help!

  1. Provide more concrete examples of the use of AI in e-petitioning.

Unfortunately, the author didn’t find to many in this area only. if the reviewer has any knowledge about any working system please advise.

Materials and Methods:

  1. Provide more detailed descriptions of the methods used to clean and process the data set.

As it is to be seen in the article, cleaning process was performed in order to delete all the data about the citizens (emails, names, addresses and so on).

  1. Discuss potential limitations of the study, particularly on the validity of the data set.

The author has reviewed the manuscript and added necessary information on the Discussions section.

The AI Model Proposed:

  1. Add background information on the use of AI in the legal field.

  1. Simplify language and provide more explanations.

The author has reviewed the manuscript and made necessary adjustments to ensure a clear understanding.

  1. Include a broader discussion on ethical considerations of using AI for petition classification.

The author has reviewed the manuscript and added necessary information on the Discussions section.

Results:

  1. Provide more information about the results obtained.

The author is kindly asking the reviewer to clarify.

  1. Discuss limitations and suggest areas for future research.

The author has reviewed the manuscript and added necessary information on the Discussions section.

Discussion:

  1. Explore the ethical implications of biased decision-making.

The author has reviewed the manuscript and added necessary information

  1. Provide more concrete examples of the use of AI in real-world applications.

The author has reviewed the manuscript and added necessary information

  1. Provide a more detailed exploration of limitations of AI systems in public administration.

The author has reviewed the manuscript and added necessary information

Conclusion:

  1. Expand upon implications for practice and discuss how findings could be applied in real-world settings.

The author has reviewed the manuscript and added necessary information

I hope these comments will be helpful to you in revising your manuscript.

The author would like to express his sincere appreciation for the reviewer's time and effort spent on providing feedback for this paper. He hopes that his responses have adequately addressed the concerns raised by the reviewer and improved the quality of the paper. Thank you again for your valuable contributions. 

Please receive our kindest regards,

Reviewer 7 Report

The paper is interesting and very relevant as well. It can be accepted. One small improvement is required in terms on its implication. I would suggest the authors(s) to include one separate section only to discuss the theoretical, practical and policy implication. Rest all fine.

All the best.

Author Response

The paper is interesting and very relevant as well. It can be accepted. One small improvement is required in terms on its implication. I would suggest the authors(s) to include one separate section only to discuss the theoretical, practical and policy implication. Rest all fine.

All the best.

The author would like to express his gratitude to the reviewer for his/her feedback and support. In response to his/her suggestions, the author has added a new section (6.3) to the article, which addresses the theoretical, practical, and policy implications of the study.

Thank you once again for your encouraging words. Best regards!

Round 2

Reviewer 1 Report

The author has correctly edited the article, no further changes are necessary.

Author Response

Thank you very much for your effort in reviewing this article!

Reviewer 2 Report

Thank you dear author for your revison made.

Our recommendations:

- Discussion section : this section is still need more improvement. Tray do add two other paragraphs to discuss more the fundings of this study.

- Literature review: we appreciate you effort ot add link site form Google open sources, read this research papers and add them in this section and also to the reference section at the end of the paper. (these papers dementrate (a)the Intelligence of the decison made in the governance context like your aim of  your study, (b) the second paper have as objective to examin with sophistic statistical method the challenged face by emarging region economy to resolve the problems of bad governance in public service like bank sector)

Hamrouni, B.; Bourouis, A.; Korichi, A.& Brahmi, M.2021. Explainable Ontology-Based Intelligent Decision Support System for Business Model Design and Sustainability, Sustainability, 13(17), 9819. https://doi.org/10.3390/su13179819

Ibtissem, M., Mohsen, B., & Jaleleddine, B.R. (2018). Quantitative relationship between corruption and development of the Tunisian stock market. Public and Municipal Finance, 7(2), 39-47. doi:10.21511/pmf.07(2).2018.04

* ABSTRACT : please, try to EDIT this sentence : ' In this paper the author will speak about'

You can use: in this study, we examin/present.............

Author Response

Thank you dear author for your revison made.

Our recommendations:

- Discussion section : this section is still need more improvement. Tray do add two other paragraphs to discuss more the fundings of this study.

Thank you for suggestions. The paragraph requested is to be found at the end of the Discussion section, before Limitation.

- Literature review: we appreciate you effort ot add link site form Google open sources, read this research papers and add them in this section and also to the reference section at the end of the paper. (these papers dementrate (a)the Intelligence of the decison made in the governance context like your aim of  your study, (b) the second paper have as objective to examin with sophistic statistical method the challenged face by emarging region economy to resolve the problems of bad governance in public service like bank sector)

Hamrouni, B.; Bourouis, A.; Korichi, A.& Brahmi, M.2021. Explainable Ontology-Based Intelligent Decision Support System for Business Model Design and Sustainability, Sustainability, 13(17), 9819. https://doi.org/10.3390/su13179819

Ibtissem, M., Mohsen, B., & Jaleleddine, B.R. (2018). Quantitative relationship between corruption and development of the Tunisian stock market. Public and Municipal Finance, 7(2), 39-47. doi:10.21511/pmf.07(2).2018.04

Thank you for suggestions. There are to be found in the paper at the beginning of the Literature review section.

* ABSTRACT : please, try to EDIT this sentence : ' In this paper the author will speak about'

You can use: in this study, we examin/present.............

Modified according with the reviewer suggestion. Thank you!

Reviewer 6 Report

Dear Authors, Our new comments in green, 

The paper focuses on the evolution of e-government from the first generation to the third generation. The author notes that the term 'e-government 2.0' is becoming less common as the focus shifts towards broader digital transformation initiatives that may include emerging technologies such as artificial intelligence, blockchain, virtual reality, and augmented reality.

The paper analyzes the challenges faced by municipalities in responding to citizen petitions, which are a core application of local democracies. The author presents an example of an e-petitions system currently in use and proposes an AI model that can deal faster and more accurately with the increased number of inputs. The author intends to promote the adoption of AI by municipalities that are still reluctant to implement it in their operations.

After having reviewed your manuscript, we have identified several areas for improvement that will help enhance the clarity and organization of your paper. Please find below a summary of my comments on each section:

Firstly, allow me to express gratitude to the reviewer for their thoughtful and encouraging feedback.

Introduction:

1.     Provide more specific research questions or hypotheses.

The author has carefully reviewed the hypotheses paragraph and added more info.

Please, indicate the line numbers of the modifications. If this is not marked, the review process gets much complicated.

2.     Organize the introduction more clearly.

One more paragraph entered the Introduction section. This will provide more context in the urban technologies sphere.

Please, indicate the line numbers of the modifications. If this is not marked, the review process gets much complicated.

3.     Use more reader-friendly language.

The author has reviewed the manuscript and made necessary adjustments to ensure a clear understanding.

Please, indicate the line numbers of the modifications. If this is not marked, the review process gets much complicated.

Literature Review:

4.     Improve organization and coherence.

The author has reviewed the manuscript and made necessary adjustments to ensure a clear understanding.

Please, indicate the line numbers of the modifications. If this is not marked, the review process gets much complicated.

5.     Include more recent research on the use of AI in e-petitioning and governance processes.

The author has included few more studies on the topic, as recommended by the reviewer. However, since the literature on this topic is extensive, the author would appreciate the reviewer's assistance in identifying relevant studies that could further enhance the article. Any recommended references would be greatly appreciated. Thank you in advance for your valuable help!

Please, indicate which are the added references. The reviewer cannot compare previous and actual versions of the manuscript for checking which are new and which are not.

6.     Provide more concrete examples of the use of AI in e-petitioning.

Unfortunately, the author didn’t find to many in this area only. if the reviewer has any knowledge about any working system please advise.

The reviewer encourages the authors to re-check again while there are several examples of this topic.

Materials and Methods:

7.     Provide more detailed descriptions of the methods used to clean and process the data set.

As it is to be seen in the article, cleaning process was performed in order to delete all the data about the citizens (emails, names, addresses and so on).

The reviewer wonder if there is no other treatment that has been done to the dataset in order to clean and process its content. It seems pretty insufficient that only deleting the personal data of the citizens could be enough.

8.     Discuss potential limitations of the study, particularly on the validity of the data set.

The author has reviewed the manuscript and added necessary information on the Discussions section.

Please, indicate the line numbers of the modifications. If this is not marked, the review process gets much complicated.

The AI Model Proposed:

9.     Add background information on the use of AI in the legal field.

No answer has been added to this point. It would be necessary to do so.

10.  Simplify language and provide more explanations.

The author has reviewed the manuscript and made necessary adjustments to ensure a clear understanding.

Please, indicate the line numbers of the modifications. If this is not marked, the review process gets much complicated.

11.  Include a broader discussion on ethical considerations of using AI for petition classification.

The author has reviewed the manuscript and added necessary information on the Discussions section.

Please, indicate the line numbers of the modifications. If this is not marked, the review process gets much complicated.

Results:

12.  Provide more information about the results obtained.

The author is kindly asking the reviewer to clarify.

In response to the authors' request for clarification, it is important to reiterate that the manuscript should provide sufficient information about the results obtained. This includes a clear and detailed presentation of data, analyses, and any other relevant findings. As reviewers, we rely on this information to evaluate the strength of the research and its implications for the field. Therefore, we respectfully request that the authors provide additional details and information about their results in order to improve the quality and impact of the manuscript.

13.  Discuss limitations and suggest areas for future research.

The author has reviewed the manuscript and added necessary information on the Discussions section.

Please, indicate the line numbers of the modifications. If this is not marked, the review process gets much complicated.

Discussion:

14.  Explore the ethical implications of biased decision-making.

The author has reviewed the manuscript and added necessary information

Please, indicate the line numbers of the modifications. If this is not marked, the review process gets much complicated.

15.  Provide more concrete examples of the use of AI in real-world applications.

The author has reviewed the manuscript and added necessary information

Please, indicate the line numbers of the modifications. If this is not marked, the review process gets much complicated.

16.  Provide a more detailed exploration of limitations of AI systems in public administration.

The author has reviewed the manuscript and added necessary information

Please, indicate the line numbers of the modifications. If this is not marked, the review process gets much complicated.

Conclusion:

17.  Expand upon implications for practice and discuss how findings could be applied in real-world settings.

The author has reviewed the manuscript and added necessary information

Please, indicate the line numbers of the modifications. If this is not marked, the review process gets much complicated.

I hope these comments will be helpful to you in revising your manuscript.

The author would like to express his sincere appreciation for the reviewer's time and effort spent on providing feedback for this paper. He hopes that his responses have adequately addressed the concerns raised by the reviewer and improved the quality of the paper. Thank you again for your valuable contributions. 

Thank you for making modifications to almost all of the comments we sent you. However, it is difficult to track any changes if they are not marked in the manuscript. We kindly request that you indicate the line numbers of all modifications made to the manuscript to make it easier for us to review. Additionally, there are some comments that have not been addressed, which is not appropriate. We respectfully ask that you please address all comments and complete any outstanding revisions necessary to improve the quality of the manuscript.

Author Response

The author expresses sympathy for the difficulties the reviewer is experiencing. Nonetheless, the author utilized the “Track Changes” feature in MS Word (as mentioned by the Managing Editor in the journal publishing recommendations - (II) ‘Any revisions to the manuscript should be marked up using the “Track Changes” function if you are using MS Word/LaTeX, such that any changes can be easily viewed by the editors and reviewers’, which ensures that all changes are fully visible (the other six reviewers were able to view them). It is possible that this particular reviewer did not download the updated version of the paper. Here are the author will mention the lines as requested.

Introduction:

  1. Provide more specific research questions or hypotheses.

The author has carefully reviewed the hypotheses paragraph and added more info.

Please, indicate the line numbers of the modifications. If this is not marked, the review process gets much complicated.

L: 99-102.

  1. Organize the introduction more clearly.

One more paragraph entered the Introduction section. This will provide more context in the urban technologies sphere.

Please, indicate the line numbers of the modifications. If this is not marked, the review process gets much complicated.

L:74-85.

  1. Use more reader-friendly language.

The author has reviewed the manuscript and made necessary adjustments to ensure a clear understanding.

Please, indicate the line numbers of the modifications. If this is not marked, the review process gets much complicated.

At this point the reviewer was very vague. The author is kindly asking the reviewer to indicate the line numbers where the author should use a more ‘reader-friendly language’.

Literature Review:

  1. Improve organization and coherence.

The author has reviewed the manuscript and made necessary adjustments to ensure a clear understanding.

Please, indicate the line numbers of the modifications. If this is not marked, the review process gets much complicated.

L: 15-19, 21-22, 74-85, 99-102, 111-113, 119-121, 156-162, 186-191, 215 in regard with the table above, 242-247, 290, 294-295, 335, 347-348, 407, 410-413, 474, 493-510, 516-517, 550-572, and also in Appendix A.

  1. Include more recent research on the use of AI in e-petitioning and governance processes.

The author has included few more studies on the topic, as recommended by the reviewer. However, since the literature on this topic is extensive, the author would appreciate the reviewer's assistance in identifying relevant studies that could further enhance the article. Any recommended references would be greatly appreciated. Thank you in advance for your valuable help!

Please, indicate which are the added references. The reviewer cannot compare previous and actual versions of the manuscript for checking which are new and which are not.

L: 8, 9, 17, 29, 44, 52, 55, 56, 57, 58, 59, 60, 77, 78, 79, 80

  1. Provide more concrete examples of the use of AI in e-petitioning.

Unfortunately, the author didn’t find to many in this area only. if the reviewer has any knowledge about any working system please advise.

The reviewer encourages the authors to re-check again while there are several examples of this topic.

 The author was adding two more relevant articles for the study (L111-113 [17] and L119-121 [29]). However, the author is kindly asking the reviewer to provide few more examples.

Materials and Methods:

  1. Provide more detailed descriptions of the methods used to clean and process the data set.

As it is to be seen in the article, cleaning process was performed in order to delete all the data about the citizens (emails, names, addresses and so on).

The reviewer wonder if there is no other treatment that has been done to the dataset in order to clean and process its content. It seems pretty insufficient that only deleting the personal data of the citizens could be enough.

There was also operation of cleaning the offensive language and some other techniques that are now to be seen in the document. Since the paper is not a technical one, the author was not seeing any reason to focus on this. However, there were added as requested (L229-237)

  1. Discuss potential limitations of the study, particularly on the validity of the data set.

The author has reviewed the manuscript and added necessary information on the Discussions section.

Please, indicate the line numbers of the modifications. If this is not marked, the review process gets much complicated.

L: 493-510, 516-517, 550-572. Meanwhile kindly take into consideration that a dedicated section for limitation is to be found in the article. The author sees no importance to add it in the Material and methods section.

The AI Model Proposed:

  1. Add background information on the use of AI in the legal field.

No answer has been added to this point. It would be necessary to do so.

Kindly revisit the lines: 259-281.

  1. Simplify language and provide more explanations.

The author has reviewed the manuscript and made necessary adjustments to ensure a clear understanding.

Please, indicate the line numbers of the modifications. If this is not marked, the review process gets much complicated.

At this point the reviewer was very vague. The author is kindly asking the reviewer to indicate the line numbers where the author should ‘simplify language and provide more explanations.

  1. Include a broader discussion on ethical considerations of using AI for petition classification.

The author has reviewed the manuscript and added necessary information on the Discussions section.

Please, indicate the line numbers of the modifications. If this is not marked, the review process gets much complicated.

Please revisit the Limitation sub-section: L512-529. There are ethical concerns addressed.

Results:

  1. Provide more information about the results obtained.

The author is kindly asking the reviewer to clarify.

In response to the authors' request for clarification, it is important to reiterate that the manuscript should provide sufficient information about the results obtained. This includes a clear and detailed presentation of data, analyses, and any other relevant findings. As reviewers, we rely on this information to evaluate the strength of the research and its implications for the field. Therefore, we respectfully request that the authors provide additional details and information about their results in order to improve the quality and impact of the manuscript.

Samples of data used have been added to the portal. Kindly access them. However, please note that the data attached to my response is not intended for public use, and should not be used or published outside of the scope of this study. Thank you for understanding.

  1. Discuss limitations and suggest areas for future research.

The author has reviewed the manuscript and added necessary information on the Discussions section.

Please, indicate the line numbers of the modifications. If this is not marked, the review process gets much complicated.

L: 493-510, 516-517, 550-572.

Discussion:

  1. Explore the ethical implications of biased decision-making.

The author has reviewed the manuscript and added necessary information

Please, indicate the line numbers of the modifications. If this is not marked, the review process gets much complicated.

Please revisit the Limitation sub-section: L512-529. There are ethical concerns addressed.

  1. Provide more concrete examples of the use of AI in real-world applications.

The author has reviewed the manuscript and added necessary information

Please, indicate the line numbers of the modifications. If this is not marked, the review process gets much complicated.

The author did not understand what the ‘real-world applications’ means for the reviewer. The author was using “real-world [without the word <applications>]” to emphasize the fact that the visual representation was made on the training set and only to provide visual representation on how the machines are distinguishing different actions among each other.

  1. Provide a more detailed exploration of limitations of AI systems in public administration.

The author has reviewed the manuscript and added necessary information

Please, indicate the line numbers of the modifications. If this is not marked, the review process gets much complicated.

Please revisit the Limitation sub-section: L512-529. There are ethical concerns addressed.

Conclusion:

  1. Expand upon implications for practice and discuss how findings could be applied in real-world settings.

The author has reviewed the manuscript and added necessary information

Please, indicate the line numbers of the modifications. If this is not marked, the review process gets much complicated.

Findings are to be found at lines: 500-510 – kindly revisit.

Round 3

Reviewer 6 Report

Dear Authors, Our new comments in green, 

The paper focuses on the evolution of e-government from the first generation to the third generation. The author notes that the term 'e-government 2.0' is becoming less common as the focus shifts towards broader digital transformation initiatives that may include emerging technologies such as artificial intelligence, blockchain, virtual reality, and augmented reality.

The paper analyzes the challenges faced by municipalities in responding to citizen petitions, which are a core application of local democracies. The author presents an example of an e-petitions system currently in use and proposes an AI model that can deal faster and more accurately with the increased number of inputs. The author intends to promote the adoption of AI by municipalities that are still reluctant to implement it in their operations.

After having reviewed your manuscript, we have identified several areas for improvement that will help enhance the clarity and organization of your paper. Please find below a summary of my comments on each section:

Firstly, allow me to express gratitude to the reviewer for their thoughtful and encouraging feedback.

The author expresses sympathy for the difficulties the reviewer is experiencing. Nonetheless, the author utilized the “Track Changes” feature in MS Word (as mentioned by the Managing Editor in the journal publishing recommendations - (II) ‘Any revisions to the manuscript should be marked up using the “Track Changes” function if you are using MS Word/LaTeX, such that any changes can be easily viewed by the editors and reviewers’, which ensures that all changes are fully visible (the other six reviewers were able to view them). It is possible that this particular reviewer did not download the updated version of the paper. Here are the author will mention the lines as requested.

Introduction:

1.     Provide more specific research questions or hypotheses.

The author has carefully reviewed the hypotheses paragraph and added more info.

Please, indicate the line numbers of the modifications. If this is not marked, the review process gets much complicated.

L: 99-102.

Thank you for indicating the line numbers. But the second sentence is not a specification of the first aim of the article which means that it still needs some modifications.

2.     Organize the introduction more clearly.

One more paragraph entered the Introduction section. This will provide more context in the urban technologies sphere.

Please, indicate the line numbers of the modifications. If this is not marked, the review process gets much complicated.

L:74-85.

These lines don’t organize the Introduction at all. And, it presents some mistakes in the references cited.

3.     Use more reader-friendly language.

The author has reviewed the manuscript and made necessary adjustments to ensure a clear understanding.

Please, indicate the line numbers of the modifications. If this is not marked, the review process gets much complicated.

At this point the reviewer was very vague. The author is kindly asking the reviewer to indicate the line numbers where the author should use a more ‘reader-friendly language’.

There are several lines where the text should be re-worded. According to comment number 2, it needs to be reorganized so it is recommended to recheck the text from this point of view.

Literature Review:

4.     Improve organization and coherence.

The author has reviewed the manuscript and made necessary adjustments to ensure a clear understanding.

Please, indicate the line numbers of the modifications. If this is not marked, the review process gets much complicated.

L: 15-19, 21-22, 74-85, 99-102, 111-113, 119-121, 156-162, 186-191, 215 in regard with the table above, 242-247, 290, 294-295, 335, 347-348, 407, 410-413, 474, 493-510, 516-517, 550-572, and also in Appendix A.

Lines 15-19 indicate the objective of the study and lines 99-102 the purpose is totally different. Lines 20-22 have some grammar mistakes. Lines 74-85 have some references mistakes. Lines 111-113 are not clear and present some grammar mistakes. Lines 242-248 present a centered format, please change it.  Etc. etc… please check them all again.

5.     Include more recent research on the use of AI in e-petitioning and governance processes.

The author has included few more studies on the topic, as recommended by the reviewer. However, since the literature on this topic is extensive, the author would appreciate the reviewer's assistance in identifying relevant studies that could further enhance the article. Any recommended references would be greatly appreciated. Thank you in advance for your valuable help!

Please, indicate which are the added references. The reviewer cannot compare previous and actual versions of the manuscript for checking which are new and which are not.

L: 8, 9, 17, 29, 44, 52, 55, 56, 57, 58, 59, 60, 77, 78, 79, 80

 6.     Provide more concrete examples of the use of AI in e-petitioning.

Unfortunately, the author didn’t find to many in this area only. if the reviewer has any knowledge about any working system please advise.

The reviewer encourages the authors to re-check again while there are several examples of this topic.

 The author was adding two more relevant articles for the study (L111-113 [17] and L119-121 [29]). However, the author is kindly asking the reviewer to provide few more examples.

Check our comment number 4 for lines 111-113.

Materials and Methods:

7.     Provide more detailed descriptions of the methods used to clean and process the data set.

As it is to be seen in the article, cleaning process was performed in order to delete all the data about the citizens (emails, names, addresses and so on).

The reviewer wonder if there is no other treatment that has been done to the dataset in order to clean and process its content. It seems pretty insufficient that only deleting the personal data of the citizens could be enough.

There was also operation of cleaning the offensive language and some other techniques that are now to be seen in the document. Since the paper is not a technical one, the author was not seeing any reason to focus on this. However, there were added as requested (L229-237)

8.     Discuss potential limitations of the study, particularly on the validity of the data set.

The author has reviewed the manuscript and added necessary information on the Discussions section.

Please, indicate the line numbers of the modifications. If this is not marked, the review process gets much complicated.

L: 493-510, 516-517, 550-572. Meanwhile kindly take into consideration that a dedicated section for limitation is to be found in the article. The author sees no importance to add it in the Material and methods section.

The AI Model Proposed:

9.     Add background information on the use of AI in the legal field.

No answer has been added to this point. It would be necessary to do so.

Kindly revisit the lines: 259-281.

10.  Simplify language and provide more explanations.

The author has reviewed the manuscript and made necessary adjustments to ensure a clear understanding.

Please, indicate the line numbers of the modifications. If this is not marked, the review process gets much complicated.

At this point the reviewer was very vague. The author is kindly asking the reviewer to indicate the line numbers where the author should ‘simplify language and provide more explanations.

11.  Include a broader discussion on ethical considerations of using AI for petition classification.

The author has reviewed the manuscript and added necessary information on the Discussions section.

Please, indicate the line numbers of the modifications. If this is not marked, the review process gets much complicated.

Please revisit the Limitation sub-section: L512-529. There are ethical concerns addressed.

Results:

12.  Provide more information about the results obtained.

The author is kindly asking the reviewer to clarify.

In response to the authors' request for clarification, it is important to reiterate that the manuscript should provide sufficient information about the results obtained. This includes a clear and detailed presentation of data, analyses, and any other relevant findings. As reviewers, we rely on this information to evaluate the strength of the research and its implications for the field. Therefore, we respectfully request that the authors provide additional details and information about their results in order to improve the quality and impact of the manuscript.

Samples of data used have been added to the portal. Kindly access them. However, please note that the data attached to my response is not intended for public use, and should not be used or published outside of the scope of this study. Thank you for understanding.

13.  Discuss limitations and suggest areas for future research.

The author has reviewed the manuscript and added necessary information on the Discussions section.

Please, indicate the line numbers of the modifications. If this is not marked, the review process gets much complicated.

L: 493-510, 516-517, 550-572.

Is it possible that these paragraphs are together in the text?

Discussion:

14.  Explore the ethical implications of biased decision-making.

The author has reviewed the manuscript and added necessary information

Please, indicate the line numbers of the modifications. If this is not marked, the review process gets much complicated.

Please revisit the Limitation sub-section: L512-529. There are ethical concerns addressed.

15.  Provide more concrete examples of the use of AI in real-world applications.

The author has reviewed the manuscript and added necessary information

Please, indicate the line numbers of the modifications. If this is not marked, the review process gets much complicated.

The author did not understand what the ‘real-world applications’ means for the reviewer. The author was using “real-world [without the word <applications>]” to emphasize the fact that the visual representation was made on the training set and only to provide visual representation on how the machines are distinguishing different actions among each other.

Could you detail your answer more? It is not very clear.

16.  Provide a more detailed exploration of limitations of AI systems in public administration.

The author has reviewed the manuscript and added necessary information

Please, indicate the line numbers of the modifications. If this is not marked, the review process gets much complicated.

Please revisit the Limitation sub-section: L512-529. There are ethical concerns addressed.

Conclusion:

17.  Expand upon implications for practice and discuss how findings could be applied in real-world settings.

The author has reviewed the manuscript and added necessary information

Please, indicate the line numbers of the modifications. If this is not marked, the review process gets much complicated.

Findings are to be found at lines: 500-510 – kindly revisit.

Are these findings in line with the aim of the study? It doesn´t seem to be so.

I hope these comments will be helpful to you in revising your manuscript.

The author would like to express his sincere appreciation for the reviewer's time and effort spent on providing feedback for this paper. He hopes that his responses have adequately addressed the concerns raised by the reviewer and improved the quality of the paper. Thank you again for your valuable contributions. 

Thank you for making modifications to almost all of the comments we sent you. However, it is difficult to track any changes if they are not marked in the manuscript. We kindly request that you indicate the line numbers of all modifications made to the manuscript to make it easier for us to review. Additionally, there are some comments that have not been addressed, which is not appropriate. We respectfully ask that you please address all comments and complete any outstanding revisions necessary to improve the quality of the manuscript.

Thank you very much for your comments. We would recommend to recheck all the text again and improve it. It still presents different mistakes that should be addressed before being considered for publication.

Author Response

Introduction:

  1. Provide more specific research questions or hypotheses.

The author has carefully reviewed the hypotheses paragraph and added more info.

Please, indicate the line numbers of the modifications. If this is not marked, the review process gets much complicated.

L: 99-102.

Thank you for indicating the line numbers. But the second sentence is not a specification of the first aim of the article which means that it still needs some modifications.

Modifications were made (L100-L103) as the reviewer was asking.

  1. Organize the introduction more clearly.

One more paragraph entered the Introduction section. This will provide more context in the urban technologies sphere.

Please, indicate the line numbers of the modifications. If this is not marked, the review process gets much complicated.

L:74-85.

These lines don’t organize the Introduction at all. And, it presents some mistakes in the references cited.

All references were made by the use of References feature in MS Word, however, the author revisited all and made the proper corrections all over the article. Also, the Introduction was reorganized.

  1. Use more reader-friendly language.

The author has reviewed the manuscript and made necessary adjustments to ensure a clear understanding.

Please, indicate the line numbers of the modifications. If this is not marked, the review process gets much complicated.

At this point the reviewer was very vague. The author is kindly asking the reviewer to indicate the line numbers where the author should use a more ‘reader-friendly language’.

There are several lines where the text should be re-worded. According to comment number 2, it needs to be reorganized so it is recommended to recheck the text from this point of view.

 Paragraphs ware reorder as suggested.

Literature Review:

  1. Improve organization and coherence.

The author has reviewed the manuscript and made necessary adjustments to ensure a clear understanding.

Please, indicate the line numbers of the modifications. If this is not marked, the review process gets much complicated.

L: 15-19, 21-22, 74-85, 99-102, 111-113, 119-121, 156-162, 186-191, 215 in regard with the table above, 242-247, 290, 294-295, 335, 347-348, 407, 410-413, 474, 493-510, 516-517, 550-572, and also in Appendix A.

Lines 15-19 indicate the objective of the study and lines 99-102 the purpose is totally different. Lines 20-22 have some grammar mistakes. Lines 74-85 have some references mistakes. Lines 111-113 are not clear and present some grammar mistakes. Lines 242-248 present a centered format, please change it.  Etc. etc… please check them all again.

Corrected as requested: Objective as in the abstract are to be seen in the introduction. Grammar was also rechecked. References were corrected all over the article. Grammar mistakes were also corrected (L111-113 are now 135-137 if MDPI conversion from doc to pdf will keep same setup). In regard with the centered lines – it is just a conversion error output, the doc file uploaded by the author is in good conditions.

  1. Include more recent research on the use of AI in e-petitioning and governance processes.

The author has included few more studies on the topic, as recommended by the reviewer. However, since the literature on this topic is extensive, the author would appreciate the reviewer's assistance in identifying relevant studies that could further enhance the article. Any recommended references would be greatly appreciated. Thank you in advance for your valuable help!

Please, indicate which are the added references. The reviewer cannot compare previous and actual versions of the manuscript for checking which are new and which are not.

L: 8, 9, 17, 29, 44, 52, 55, 56, 57, 58, 59, 60, 77, 78, 79, 80

  1. Provide more concrete examples of the use of AI in e-petitioning.

Unfortunately, the author didn’t find to many in this area only. if the reviewer has any knowledge about any working system please advise.

The reviewer encourages the authors to re-check again while there are several examples of this topic.

 The author was adding two more relevant articles for the study (L111-113 [17] and L119-121 [29]). However, the author is kindly asking the reviewer to provide few more examples.

Check our comment number 4 for lines 111-113.

Rechecked as requested.

Materials and Methods:

  1. Provide more detailed descriptions of the methods used to clean and process the data set.

As it is to be seen in the article, cleaning process was performed in order to delete all the data about the citizens (emails, names, addresses and so on).

The reviewer wonder if there is no other treatment that has been done to the dataset in order to clean and process its content. It seems pretty insufficient that only deleting the personal data of the citizens could be enough.

There was also operation of cleaning the offensive language and some other techniques that are now to be seen in the document. Since the paper is not a technical one, the author was not seeing any reason to focus on this. However, there were added as requested (L229-237)

  1. Discuss potential limitations of the study, particularly on the validity of the data set.

The author has reviewed the manuscript and added necessary information on the Discussions section.

Please, indicate the line numbers of the modifications. If this is not marked, the review process gets much complicated.

L: 493-510, 516-517, 550-572. Meanwhile kindly take into consideration that a dedicated section for limitation is to be found in the article. The author sees no importance to add it in the Material and methods section.

The AI Model Proposed:

  1. Add background information on the use of AI in the legal field.

No answer has been added to this point. It would be necessary to do so.

Kindly revisit the lines: 259-281.

  1. Simplify language and provide more explanations.

The author has reviewed the manuscript and made necessary adjustments to ensure a clear understanding.

Please, indicate the line numbers of the modifications. If this is not marked, the review process gets much complicated.

At this point the reviewer was very vague. The author is kindly asking the reviewer to indicate the line numbers where the author should ‘simplify language and provide more explanations.

  1. Include a broader discussion on ethical considerations of using AI for petition classification.

The author has reviewed the manuscript and added necessary information on the Discussions section.

Please, indicate the line numbers of the modifications. If this is not marked, the review process gets much complicated.

Please revisit the Limitation sub-section: L512-529. There are ethical concerns addressed.

Results:

  1. Provide more information about the results obtained.

The author is kindly asking the reviewer to clarify.

In response to the authors' request for clarification, it is important to reiterate that the manuscript should provide sufficient information about the results obtained. This includes a clear and detailed presentation of data, analyses, and any other relevant findings. As reviewers, we rely on this information to evaluate the strength of the research and its implications for the field. Therefore, we respectfully request that the authors provide additional details and information about their results in order to improve the quality and impact of the manuscript.

Samples of data used have been added to the portal. Kindly access them. However, please note that the data attached to my response is not intended for public use, and should not be used or published outside of the scope of this study. Thank you for understanding.

  1. Discuss limitations and suggest areas for future research.

The author has reviewed the manuscript and added necessary information on the Discussions section.

Please, indicate the line numbers of the modifications. If this is not marked, the review process gets much complicated.

L: 493-510, 516-517, 550-572.

Is it possible that these paragraphs are together in the text?

While the first paragraphs are describing different other utilities for the proposed model (being able to understand text it can be used for various purposes, such as platforms where users are giving feedback for specific products or services - not necessarily governmental), the last ones from the list mentioned above, are presenting the implications from a theoretical and practical perspective; implications that are presenting after the readers will priorly read the Limitation and Future work subsections.

Discussion:

  1. Explore the ethical implications of biased decision-making.

The author has reviewed the manuscript and added necessary information

Please, indicate the line numbers of the modifications. If this is not marked, the review process gets much complicated.

Please revisit the Limitation sub-section: L512-529. There are ethical concerns addressed.

  1. Provide more concrete examples of the use of AI in real-world applications.

The author has reviewed the manuscript and added necessary information

Please, indicate the line numbers of the modifications. If this is not marked, the review process gets much complicated.

The author did not understand what the ‘real-world applications’ means for the reviewer. The author was using “real-world [without the word <applications>]” to emphasize the fact that the visual representation was made on the training set and only to provide visual representation on how the machines are distinguishing different actions among each other.

Could you detail your answer more? It is not very clear.

Happily. Out of the total number of petitions, only 10% were used for fine-tunning the model. Once this is done, in machine learning engineering, the model is ready to be tested “in the real world” which means will be able to act live on new inputs (e-petitions in this case), just as a human operator – however, supervised by one, based on the arguments as discussed on the article (and of course on other scientific articles as presented in the reference section). Those new inputs won’t be already filtered (as mentioned in the text), as was in the case of those received from the municipality. The author can, of course, provide many examples of AI apps working today: from Facebook automatic filtering of hate-speech, to Amazon or Netflix recommendation system, Google search-bar autocomplete suggestions and all the way to self-driving cars that are able to detect live objects in the nearby environment. All of those (and many other) apps were trained on inputs previously owned by the companies. Depending on how robust the system needs to be the training set could vary in size. For example, Amazon or Netflix recommendations don’t need to be as accurate as a self-driving car in understanding the environment, therefore, there is no need for Amazon systems to “read” all the customer complains for a product to understand that the consumers don’t like one in particular, while for a Tesla would be a good idea to understand how pedestrians look like in different light, dressed in different colors and so on. For the purpose of these paper there is no reason to test the model on the whole number of petitions. The words used by citizens are limited to few thousands (and the whole dictionary of words in the training set was uploaded on the MDPI platform as stated in one of my previous answers), and the value of connections among them tends to be the same after “reading/learning/training” a couple of sentences – as an example, there is no chance for a citizen to complain about the presence of a lion on the streets (so the value of connection between the word “lion” and “street” is zero) but some of them do complain about the potholes, so the value of the connection (strength) is a number that tend to be the same all over the text corpus. In addition to this learning process, the same model, trained on the training set from Brasov, could be deployed on a different municipality without any extra need for the system to read previous petitions addressed to the new municipality (not to mention that this new municipality maybe was not using any online services to collect that from the citizens therefore there is nothing to learn from it). I hope the reviewer understand now what is the meaning of the “real world” in regard to AI.

  1. Provide a more detailed exploration of limitations of AI systems in public administration.

The author has reviewed the manuscript and added necessary information

Please, indicate the line numbers of the modifications. If this is not marked, the review process gets much complicated.

Please revisit the Limitation sub-section: L512-529. There are ethical concerns addressed.

Conclusion:

  1. Expand upon implications for practice and discuss how findings could be applied in real-world settings.

The author has reviewed the manuscript and added necessary information

Please, indicate the line numbers of the modifications. If this is not marked, the review process gets much complicated.

Findings are to be found at lines: 500-510 – kindly revisit.

Are these findings in line with the aim of the study? It doesn´t seem to be so.

The real value of such a system relies on the speed and also accuracy (no human is capable of reading 130 petitions in 2-16 seconds keeping the accuracy to minimum 80%, and if they would, they won’t be able to work 24/7) – as mentioned in the Results and Discussion sections of the article. However, the model is effective not only for e-petitions, but also for other text-input platforms, such as complaints in e-commerce or viewers comments about movies on different platforms, feedback from students in regard to a course material and so on.

I hope these comments will be helpful to you in revising your manuscript.

The author would like to express his sincere appreciation for the reviewer's time and effort spent on providing feedback for this paper. He hopes that his responses have adequately addressed the concerns raised by the reviewer and improved the quality of the paper. Thank you again for your valuable contributions. 

Thank you for making modifications to almost all of the comments we sent you. However, it is difficult to track any changes if they are not marked in the manuscript. We kindly request that you indicate the line numbers of all modifications made to the manuscript to make it easier for us to review. Additionally, there are some comments that have not been addressed, which is not appropriate. We respectfully ask that you please address all comments and complete any outstanding revisions necessary to improve the quality of the manuscript.

Thank you very much for your comments. We would recommend to recheck all the text again and improve it. It still presents different mistakes that should be addressed before being considered for publication.

I appreciate your generosity and the time you took to review the article. Hopefully, the reviewer is satisfied by the content as it is now.
